# Systematic attribution of heatwaves to the emissions of carbon majors

Yann Quilcaille[1 ✉], Lukas Gudmundsson[1], Dominik L. Schumacher[1], Thomas Gasser[2], Richard Heede[3], Corina Heri[4], Quentin Lejeune[5], Shruti Nath[6], Philippe Naveau[7], Wim Thiery[8], Carl-Friedrich Schleussner[2,9] & Sonia I. Seneviratne[1]

Extreme event attribution assesses how climate change affected climate extremes, but typically focuses on single events[1–4]. Furthermore, these attributions rarely quantify the extent to which anthropogenic actors have contributed to these events[5,6]. Here we show that climate change made 213 historical heatwaves reported over 2000–2023 more likely and more intense, to which each of the 180 carbon majors (fossil fuel and cement producers) substantially contributed. This work relies on the expansion of a well-established event-based framework[1]. Owing to global warming since 1850–1900, the median of the heatwaves during 2000–2009 became about 20 times more likely, and about 200 times more likely during 2010–2019. Overall, one-quarter of these events were virtually impossible without climate change. The emissions of the carbon majors contribute to half the increase in heatwave intensity since 1850–1900. Depending on the carbon major, their individual contribution is high enough to enable the occurrence of 16–53 heatwaves that would have been virtually impossible in a preindustrial climate. We, therefore, establish that the influence of climate change on heatwaves has increased, and that all carbon majors, even the smaller ones, contributed substantially to the occurrence of heatwaves. Our results contribute to filling the evidentiary gap to establish accountability of historical climate extremes[7,8].

Human-induced global warming not only causes long-term changes of state variables, energy and water fluxes in the Earth system but also manifests through climate extremes[9]. Every region of the world exhibits changes in intensity and frequency of extreme weather and climate events[10,11], and events that were near impossible in the past are now occurring[10,12]. To assess the extent of contribution of climate change to these events, the field of extreme event attribution (EEA) has developed over the past years, through approaches promoted by the World Weather Attribution (WWA) initiative[1] and other methods[2–4].

These approaches have been used to study many individual extreme events[13], often showing an important contribution of climate change. However, to our knowledge, there is no framework to systematically and collectively conduct attribution exercises on a set of events identified in past records[14], implying that impactful extreme events may still not be assessed.

Moreover, EEA studies typically attribute events to climate change, but rarely to its causes[5,6]. Extending EEA to source attribution provides the quantification of the causal chain from emitters to climate extremes. It has been proven unambiguously that anthropogenic activities are largely responsible for climate change, and that combustion of fossil fuels is the main contributor[15]. Three categories of emitters may be used: countries[5], individuals[6] or businesses. In the first case, the source allocation of emissions can be based on the territorial origin of emissions produced within country borders[16,17]. Consumption-based allocations can be pursued, as well as approaches based on individual emission profiles[6,18]. Finally, emissions can also be allocated to businesses that directly profit from fossil fuel production or other high-emitting activities[19–24]. Businesses with particularly high emission profiles are referred as carbon majors, encompassing not only investor-owned companies (for example, ExxonMobil) but also state-owned companies (for example, Saudi Aramco) or nation-state producers (for example, the former Soviet Union)[19].

Here, we address both issues: the lack of systematic attribution of extreme events and the absence of quantitative analysis establishing a causal chain from individual emitters to these events. We build on an existing and widely used EEA framework[1], systematizing the approach. We assess how much climate change has contributed to 213 heatwaves reported in the international disaster database EM-DAT over 2000–2023, owing to their particularly significant impact, most of which were previously unattributed. Then, we build on existing approaches to assess contributions to climate change[5,6], extending the attribution upstream to the emitters. We assess how much the emissions of the 180 biggest carbon majors[19] contributed to global mean surface temperature and to the likelihood and severity of historical heatwaves.

[1]Institute for Atmospheric and Climate Science, Department of Environmental Systems Science, ETH Zurich, Zurich, Switzerland. [2]International Institute for Applied Systems Analysis (IIASA), Laxenburg, Austria. [3]Climate Accountability Institute, Snowmass, CO, USA. [4]Department of Public Law and Governance, Tilburg University, Tilburg, The Netherlands. [5]Climate Analytics, Berlin, Germany. [6]Atmospheric, Oceanic and Planetary Physics, Department of Physics, University of Oxford, Oxford, UK. [7]Laboratoire des Sciences du Climat et de l'Environnement, ESTIMR, CNRS-CEA-UVSQ, Gif-sur-Yvette, France. [8]Vrije Universiteit Brussel, Department of Water and Climate, Brussels, Belgium. [9]Integrative Research Institute on Transformations of Human-Environment Systems (IRI THESys) and the Geography Department, Humboldt-Universität zu Berlin, Berlin, Germany. ✉e-mail: yann.quilcaille@env.ethz.ch

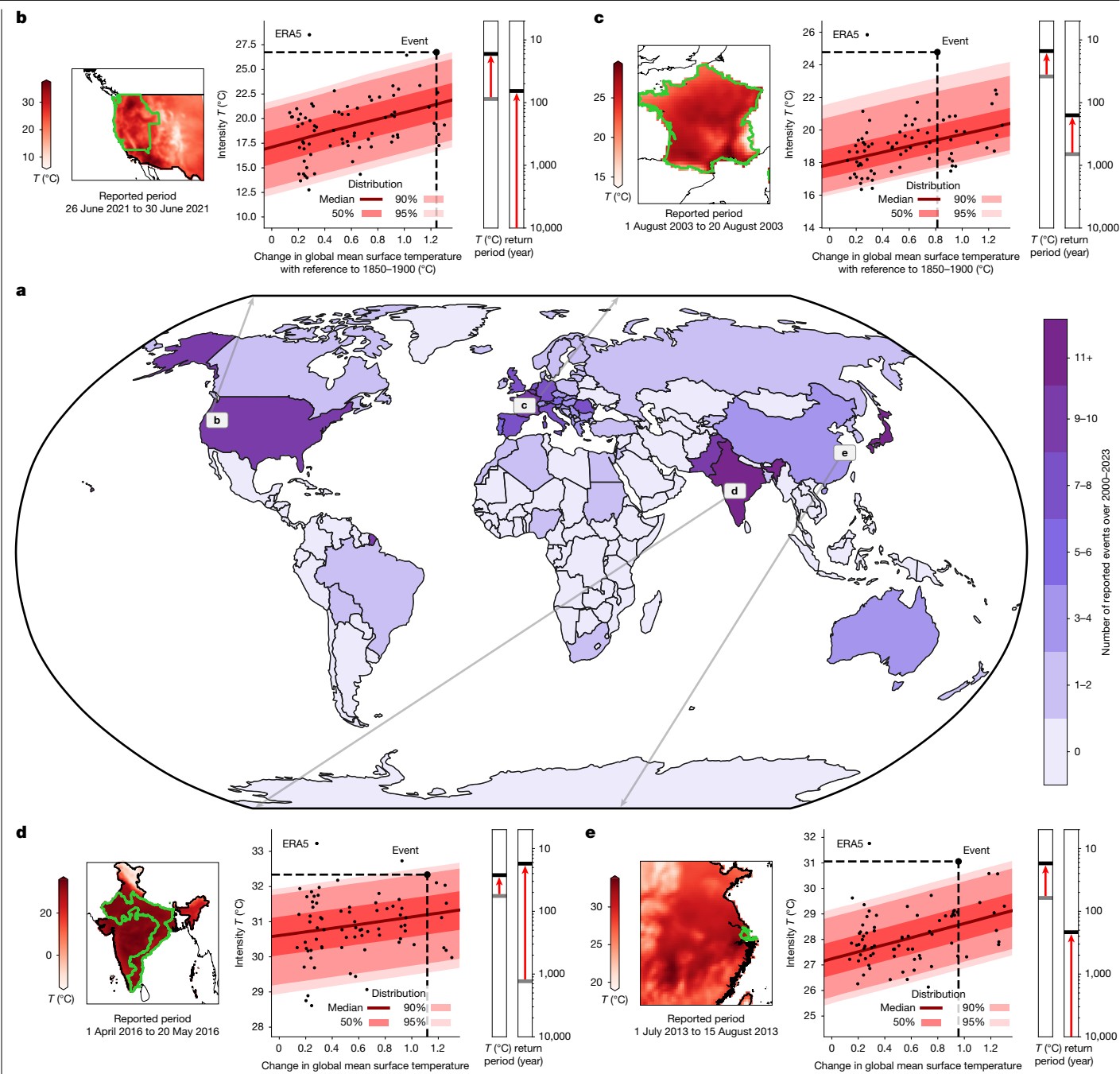

**Fig. 1 | For every reported heatwave, the contribution of climate change to the event is assessed using statistical models and multiple lines of evidence.** **a**, The number of heatwaves reported per country in EM-DAT (www.emdat.be) over 2000–2023. An EEA is performed for each of them, as shown for four examples with ERA5 (ref. 28) data. **b**, The 2021 Pacific Northwest heat dome. **c**, The 2003 heatwave in France. **d**, The 2022 Indian heatwave. **e**, The 2013 eastern China heatwave. For each example, the average temperatures during the event are mapped, with the outlines of the reported region (lime green contours).

Moreover, the intensity (average temperature $T$ (°C) during the period and the region of the event) and change in GMST (°C) are represented over 1950–2023 (black dots), with their conditional distribution represented through the median (red line) and ranges of the distribution (red shading). Finally, the change in intensity and change in return period (year) compared with the preindustrial reference period are shown for each example. Uncertainties inferred using bootstrapping are not shown here for the sake of clarity. Further details are provided in the Methods.

## Systematic attribution of heatwaves

In the EM-DAT database (www.emdat.be), 226 heatwaves are reported over 2000–2023, across 63 countries (Fig. 1a). These events were reported because of significant economic losses or casualties, a declaration of state of emergency or a call for international assistance. These societal impacts warrant their relevance for event attribution. Despite EM-DAT being the most widely used disaster database, the reporting of heatwaves across countries is highly uneven, with only nine heatwaves out of 226 reported over Africa, Latin America and the Caribbean, although these regions are also prone to heatwaves[10]. This known reporting bias in the EM-DAT database[25] calls for more complete reporting to enable a more exhaustive analysis.

Each of these heatwaves is systematically characterized and analysed in a consistent framework, following the method promoted by the WWA initiative[1]. This method is shown with the Pacific Northwest heatwave

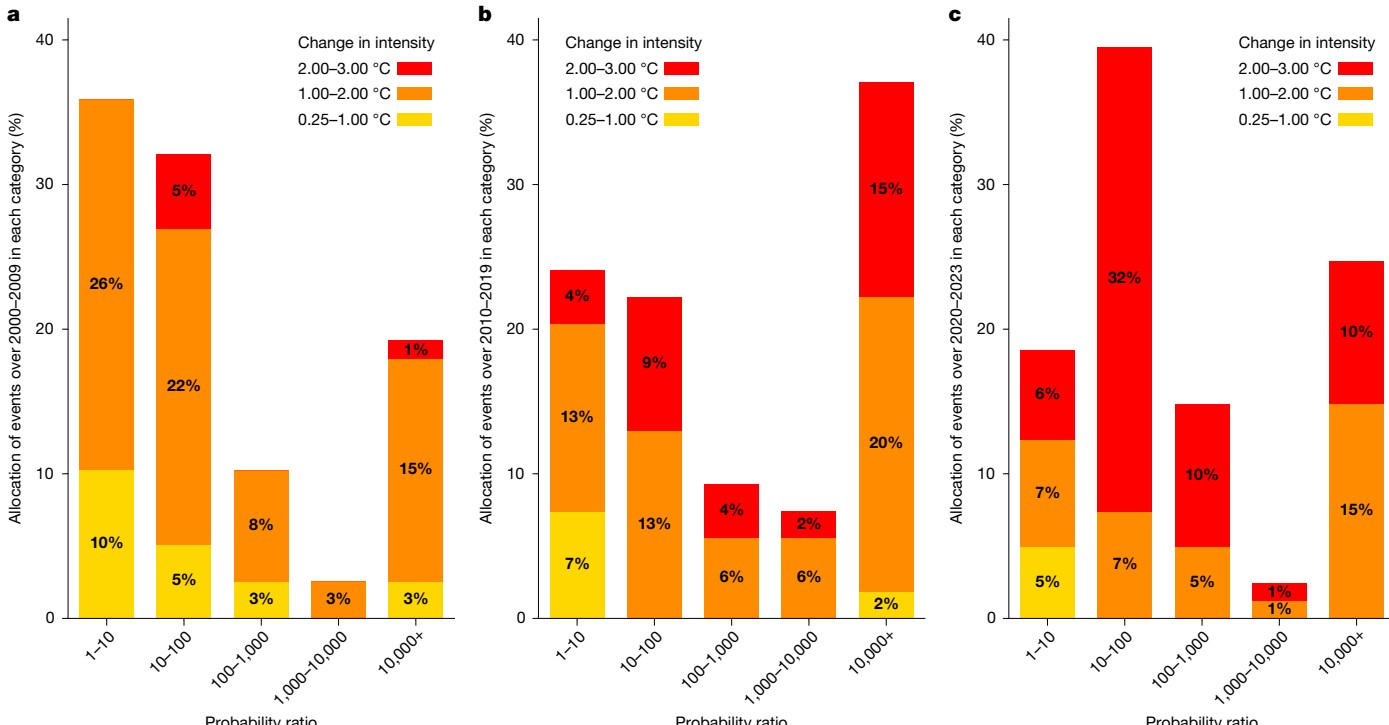

**Fig. 2 | Increasing contribution of climate change to 213 heatwaves over time.** Each heatwave is allocated a category depending on its change in intensity (colour) and its probability ratio (vertical bars in per cent) with reference to 1850–1900. **a**–**c**, Events are categorized based on the year of the event: 78 heatwaves attributed over 2000–2009 (**a**), 54 heatwaves attributed over 2010–2019 (**b**) and 81 heatwaves attributed over 2020–2023 (**c**). Median results are shown here. Further details on the attribution of each heatwave event are provided in the Methods, and all results are provided in the Supplementary Information.

of 2021 as reported for the United States (Fig. 1b), thus without British Columbia in Canada. This event was reported in Oregon, Washington, Northern California, Idaho and Western Nevada over 26–30 June 2021 (www.emdat.be). Usually, EEA defines the heatwave with a box surrounding the region[1]. Here the heatwave is defined using the exact spatial characterization in the EM-DAT database, as it represents how the disaster was experienced by the local populations[1] (Fig. 1b). Daily temperatures are averaged over this period and region for every year of available observations. Although many indicators could be used to characterize the heatwave[26], the choice of the average over the period is motivated by its relevance for the reported impact rather than its meteorological rarity (Methods). Following the method by WWA[1] and justified by the extreme value theory[27], a statistical relationship can be inferred that links the probability distribution of the event to the change in global mean surface temperature[1] (GMST) (Fig. 1b). This relationship allows us to calculate the probability and the intensity of the heatwave, both under observed conditions with climate change and under the pre-industrial climate of 1850–1900 without such perturbations (Fig. 1b). Using both observation-derived estimates (ERA5 (ref. 28) and BEST (ref. 29)) and Earth system models[30], these results synthesize how climate change has affected the heatwave through a change in intensity and how many times more likely the event has become, which is called the probability ratio[1]. More details on the systematization of the WWA approach are provided in the Methods. Using only ERA5[28], the Pacific Northwest heatwave of 2021 over the United States had climate change increasing its intensity by 4.4 °C compared with that in 1850–1900, with a 95% confidence interval of 2.2–6.8 °C. Adding all other datasets[29,30] relevant for the region decreases the influence of climate change to a change in intensity of the Pacific Northwest heatwave of 2021 over the United States of 3.1 °C (1.4–5.1 °C). The median estimate indicates that climate change has also increased the probability of heatwaves by more than 10,000, and at least seven times according to the lower bound of

the confidence interval. This attribution is consistent with existing works on this heatwave: a previous work[31] found a probability ratio of at least 150 and a change in intensity of 2.0 °C (1.2– 2.8 °C), whereas another analysis[32] suggested a change in intensity of more than 2.9 °C. Although consistent, the results differ because of the characterization of the event and its very unlikely nature. The region and period are here determined through the reporting of the disaster for relevance to the impact rather than choices motivated by its meteorological rarity. Choosing the maximum temperature as an indicator over the period amplifies the extremeness of the event[32,33]. Very unlikely events such as the Pacific Northwest heatwave of 2021 are more difficult to investigate, increasing the dispersion across several analysis[34]. However, this increased dispersion does not lower the confidence in the conclusion, which is the strong influence of climate change on these unlikely events. The method described for this event is applied to the 226 heatwaves, with three other cases shown in Fig. 1c–e, with the results also consistent with available attribution studies.

Additional tests are conducted to assess the adequacy of the method for each event. The goodness of fit is assessed, validating 217 out of the 226 heatwaves, whereas the remaining nine are removed from the ensuing analysis. Furthermore, although there are strong physical justifications that GMST has a causal link to the heatwave[1,10], this statistical model does not necessarily imply statistical causation. Thus, we also infer the non-linear Granger causality[4]. For 214 out of the 217 heatwaves, we prove with more than 95% certainty that GMST is a Granger-causing indicator of the heatwave. The three other events are removed from the ensuing analysis. Finally, another heatwave is removed because of the ensuing analysis related to the carbon majors. All details on these tests are provided in the Methods.

Our analysis shows that human-induced climate change has contributed to increasing the intensity of all 213 heatwaves analysed here (Fig. 2). With reference to 1850–1900, the median estimates for the

changes in intensity range across events from +0.3 °C to +2.9 °C. The latter is the heatwave introduced in Fig. 1b, the Pacific Northwest heatwave of 2021 over the United States, whereas the heatwave with the mildest change in intensity occurred in Pakistan in June 2000. Over the study period, attributed heatwaves have become more and more intense (Fig. 2a–c). The median of the events shows that climate change has increased the intensity by 1.4 °C over 2000–2009, 1.7 °C over 2010–2019 and 2.2 °C over 2020–2023. This is consistent with GMST increasing by more than 0.2 °C per decade over the study period, and land warming faster[35].

Apart from increasing the intensity, climate change has also increased the probability of all 213 heatwaves. The lowest probability ratio is observed for the heatwave of May 2006 in India, in which the event became only 22% more likely. However, the median estimates show that climate change has made 55 heatwaves out of 213 (26%) at least 10,000 times more likely, with a 95% confidence interval ranging from 7 to 158. This probability ratio is equivalent to saying that these heatwaves would have been virtually impossible without anthropogenic influence. Over the study period, the contribution of climate change to the likelihood of these events is also increasing (Fig. 2a–c). Figure 2 shows that the probability ratios of the heatwaves are shifting from low values to higher values, although this trend is highly affected by natural variability. The median probability ratios show that, with reference to 1850–1900, climate change made heatwaves about 20 times more likely over 2000–2009, and about 200 times more likely over 2010–2019. Overall, this systematic attribution of 213 heatwaves enhances the capacity of EEA for analyses across events, proving that climate change made all events more intense and more likely, and that this influence is increasing over time with increased global warming.

## Attribution to emissions of carbon majors

EEA has been extensively used to quantify how human-induced climate change influences extreme events[13]. A growing literature has also investigated the contributions of anthropogenic actors to climate change[16,18,21]. However, the quantification of the causal chain from individual emitters to the extreme events has only been pursued in selected cases[5,6]. Here we build on established approaches while also introducing key methodological advancements in the decomposition (Methods) and developing the framework for simultaneous investigation over a large set of events. Unlike previous works that focus on emissions by countries[5] or individuals[6], we here investigate the attribution of emissions from businesses and specifically the carbon majors. Following established approaches[20–23], we assign to each carbon major the emissions associated with the full value chain of their products, including all emissions in line with established accounting and reporting standards for corporates. This modelling choice aims at filling a gap in the scientific literature and does not preclude broader reflections on emission allocations and business responsibilities[20].

The emissions from carbon majors are estimated from company production records and associated emission factors[19], leading to a dataset that provides $CO_2$ and $CH_4$ emissions for 180 carbon majors over 1854–2023 (Fig. 3a). Altogether, the emissions from these carbon majors represent 57% of the total cumulative anthropogenic $CO_2$ emissions, including land use over the 1850–2023 period[36]. When considering only the emissions from fossil fuels and cement, the emissions from these carbon majors represent 75% of the cumulative $CO_2$ emissions over 1850–2023 (ref. 36). The carbon majors have heterogeneous contributions to the $CO_2$ emissions. The 14 top carbon majors (the former Soviet Union, People's Republic of China for coal, Saudi Aramco, Gazprom, ExxonMobil, Chevron, National Iranian Oil Company, BP, Shell, India for coal, Pemex, CHN Energy, People's Republic of China for cement) represent 30% of the total cumulative anthropogenic $CO_2$

emissions, including land use, about as much as the 166 other carbon majors combined (27%). From a national perspective, 33 carbon majors are headquartered in the United States, accounting for 10% of the total $CO_2$ emissions, and 33 carbon majors are headquartered in China (12% of the total $CO_2$ emissions).

Based on the $CO_2$ and $CH_4$ emissions of the carbon majors, we compute the contributions of these carbon majors to GMST. Climate models may be used to calculate climate change over the historical period, but also counterfactual worlds, such as a world in which a given carbon major would not have emitted. The difference informs how much this single actor has warmed the Earth over time. This method has already been applied to a former version of this database using a simple impulse-response model for $CO_2$ (ref. 21). Here we use the reduced-complexity Earth system model OSCAR, for its non-linear representation of both the carbon cycle and the atmospheric chemistry of methane, as well as its capacity to integrate observational constraints to improve the robustness of the assessment[37] (Methods).

Closely aligned with the estimates based on IPCC methodologies[35], we estimate an increase in GMST of about 1.30 °C in 2023 with respect to 1850–1900 (ref. 28), of which 0.67 °C is due to the emissions of all carbon majors and 0.33 °C is due to the emissions of the 14 biggest carbon majors (Fig. 3b). The unattributed 0.63 °C is due to other actors responsible for unaccounted fossil fuel burning, agricultural and land-use activities, other industrial processes, as well as to non-attributed greenhouse gases ($N_2O$ and halogenated species) and short-lived climate forcers. For comparison, a former assessment associates 0.40 °C in 2010 with 90 carbon majors[21], whereas we find 0.48 °C with 180 carbon majors in 2010.

Knowing the contributions of the carbon majors to GMST and knowing the relationship between GMST and the heatwaves from the event attribution, we subsequently compute how each carbon major has affected each heatwave (for details, see the Methods). For each heatwave, the total effect of climate change on the intensity and probability of the event is decomposed into the contributions from individual carbon majors and the combined effect of other unidentified contributors, anthropogenic and natural.

Contributions from carbon majors to the intensities of all heatwaves range from 0 °C to 0.18 °C (Fig. 3c–k). As expected, the higher the emissions from a carbon major, the higher its contributions to the intensities of the heatwaves. The median contributions to heatwaves from the 14 top carbon majors range from 0.01 °C to 0.09 °C. The other carbon majors have lower contributions, although the 166 of them combined have about the same importance as the biggest carbon majors. We calculate the influence of carbon majors on heatwaves reported over each decade of our dataset. With reference to 1850–1900, climate change has increased the median intensity of heatwaves by 1.36 °C over 2000–2009, of which 0.44 °C is traced back to the 14 top carbon majors and 0.22 °C to the 166 others. These contributions correspond, respectively, to 32% and 16% of the overall effect of climate change. Over 2010–2019, the influence of climate change increased to 1.68 °C, with 0.47 °C (28%) from the 14 top carbon majors and 0.38 °C (22%) from the 166 others. These results show that the emissions of carbon major contributed to about half of the increase in intensity of heatwaves since preindustrial times, and that this contribution is rising.

Apart from intensities, all the carbon majors have also increased the probability of all the heatwaves. For heatwaves that climate change made only slightly more likely, or for carbon majors with much lower emissions, the contributions are limited to an increase by 10% of the preindustrial probability. However, there are heatwaves that the carbon majors have made at least 10,000 times more likely compared with preindustrial levels, and which would have otherwise been virtually impossible without anthropogenic influence. Even relatively minor shares in total emissions lead to very substantial increases in

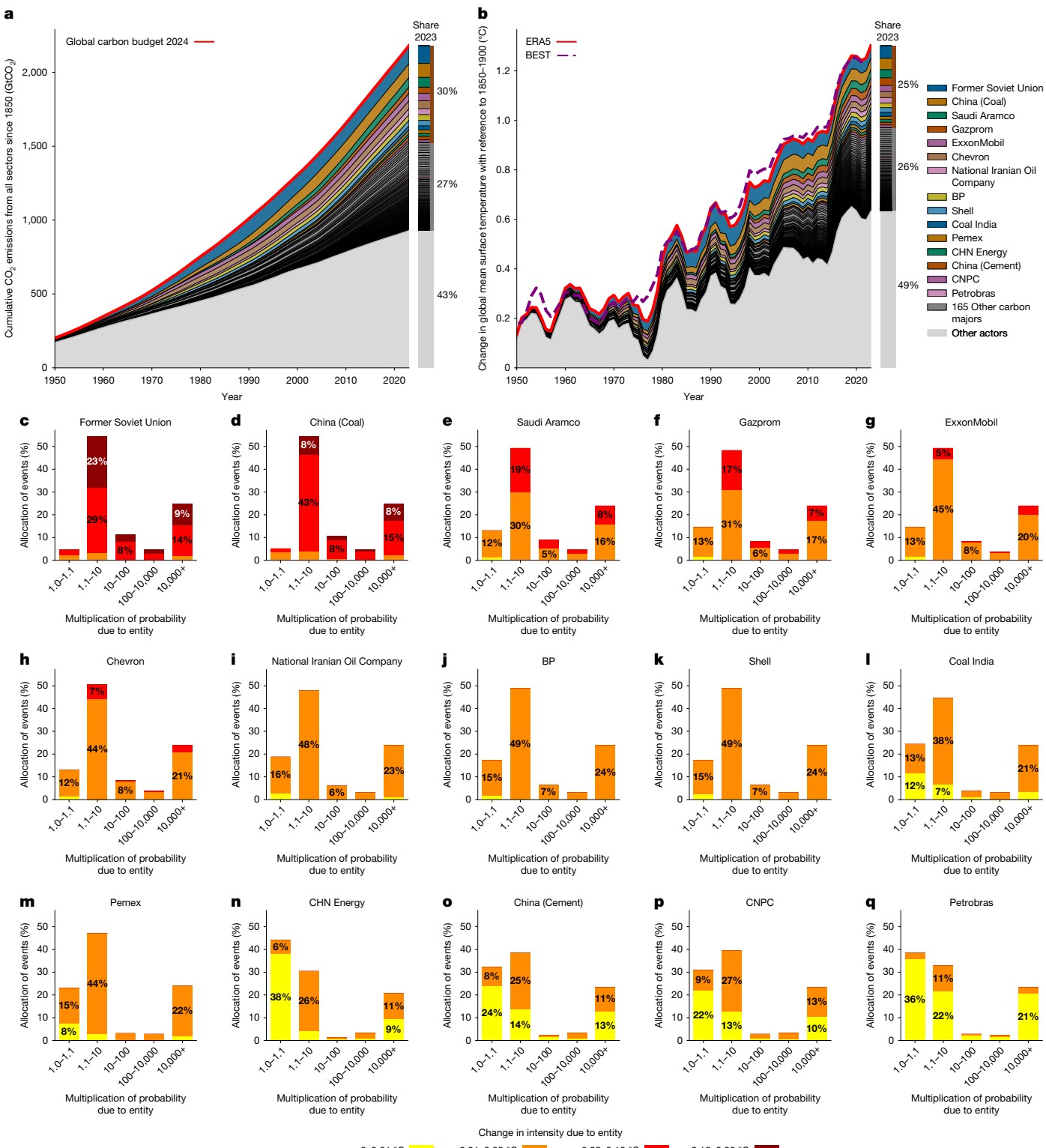

**Fig. 3 | Through their emissions, every carbon major contributes to climate change and thus to the heatwaves, even relatively smaller carbon majors.** **a,b**, Contributions of the carbon majors to the cumulative $CO_2$ emission since 1850 (all sectors) (**a**) as reported in the Carbon Majors database (https://carbonmajors.org/) and compared with the Global Carbon Budget[36] and the ensuing GMST as simulated by the OSCAR model (**b**). The GMST of ERA5[28] and BEST[29] have been rebased to 1850–1900 (ref. 9). **c–q**, Attribution of historical

heatwaves to the emissions of carbon majors for 15 selected carbon majors. In each of these panels, the 213 heatwaves are allocated into categories of contributions of the carbon majors to the change in intensity (colours) and how many times the carbon major increased the likelihood of the heatwave compared to 1850–1900 (x-axis). The results are shown through the median, but all results are provided in the Supplementary Information.

the frequency of these events. Specifically, emissions associated with the biggest emitter, the former Soviet Union, have made 53 heatwaves (25%) at least 10,000 times more likely. For the smallest carbon major by emissions, Elgaugol, this is still the case for 16 heatwaves (8%). It means that the sole emissions of these carbon majors would have rendered possible these heatwaves otherwise virtually impossible.

## Discussion

We have systematized the process of EEA, based on the widely used method promoted by the WWA initiative[1]. We achieve the analysis of 213 heatwaves, thus extending the coverage of existing event attribution studies. We validate the goodness of fit and the causality for each of these events. We show that climate change has increased the probability and intensity of all these heatwaves. Owing to the consistent protocol across all events, their meta-analysis over time shows that the extremeness of the heatwaves is rising more and more rapidly because of climate change, both in intensity and probability.

We also extend the attribution analysis upstream along the causal chain, providing a coherent attribution to individual emissions on the company level for 180 carbon majors. The contributions of the carbon majors are very heterogeneous, with 14 carbon majors (the former Soviet Union, People's Republic of China for coal, Saudi Aramco, Gazprom, ExxonMobil, Chevron, National Iranian Oil Company, BP, Shell, India for coal, Pemex, CHN Energy, People's Republic of China for cement) contributing as much as the 166 others. Considering all reporting heatwaves, we show that carbon majors represent about half of the change in intensity since 1850–1900 and that their contributions are rising, in particular the ones of the smaller carbon majors. The analysis of their contributions to the probabilities of the heatwaves shows that although the contributions scale well with their cumulative emissions, smaller carbon majors cannot be neglected. Depending on the carbon major, between 16 and 53 of these heatwaves are made possible with the sole contribution of the smaller carbon majors.

Although this assessment builds on well-established methods, there are still two limitations in this work. Although the EM-DAT is the most complete existing database for disasters, many heatwaves are still not reported, calling for a more exhaustive coverage of the events. Moreover, the contributions of the carbon majors remain incomplete. On the one hand, not all $CO_2$ and $CH_4$ emissions are covered in this database because of underreporting[19]. For instance, this database represents only 75% of the fossil fuel and cement $CO_2$ emissions reported over 1850–2023 (ref. 36). The actual contributions of the carbon majors are thus expected to be higher if all the emissions from fossil fuel and cement producers are included. On the other hand, the burning of fossil fuels can release aerosols that would have a local effect on the climate. As a whole, the aerosols emitted by the fossil fuel sector reduce their contribution by approximately 10% (ref. 38). However, attributing aerosol climate effects to individual companies would be highly challenging. The effects of aerosols on climate are local to regional, yet fossil fuels are globally traded. Furthermore, aerosol emissions from fossil fuel combustion strongly depend on the use of filter technology, which differs between regions, sectors and combustion techniques. If these challenges are overcome, it would pave the way for attributing the aerosol health effects to individual emitters. Aerosols are also harmful air pollutants, with the emissions by the use of fossil fuel causing about 5 million excess deaths per year (ref. 39). Accounting for these effects remains beyond the scope of our analysis.

Our framework could be adapted to other physical hazards, such as ocean acidity[22], sea-level rise[40], fires[41] or droughts[42]. Extending the attribution from physical hazards to societal impacts remains a challenge. We may use directly the fraction of attributable risk to deduce the fraction of the impact imputed to the actor[6], but it neglects complex and non-linear aspects related to the vulnerability and the exposure to the hazard[43]. Nevertheless, this attribution framework may be extended to heat-related mortality[44] or economic damages[24]. Finally, other top-down approaches can complement our findings[45].

These results are relevant not only in the scientific community but also for climate policy, litigation and wider efforts concerning corporate accountability[8]. Climate-related legal proceedings are proliferating, with defendants seeking compensation for losses and damages or requiring more ambitious climate actions from corporations and nations[8]. However, the scientific evidence backing the claims is often lagging behind the state of the art in climate science, thus failing to adequately draw causality links[7]. Although this work aims at filling in scientific gaps, the results also fill in evidentiary gaps. This systematic attribution improves the coverage in extreme events, thus reinforcing the potential of attribution science for climate litigation[7,43]. Furthermore, if the fact that fossil fuels are the main driver of climate change has been unambiguously established[15,36], as acknowledged by the carbon majors themselves[46–48], proving and quantifying the causality from the emitters to the events provides important new resources to assess legal responsibilities. Further strengthening of the links between climate scientists, legal scholars and practitioners is beneficial to ensure that the overwhelming scientific literature is correctly accounted for[49].

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

## Methods

### Definition of events

From the EM-DAT database, we select the events only after 2000, because the reporting is more complete after this date[25,50], and since climate change has been shown to exert increasing influence on extremes over this period[51].

The locations reported for the events in the EM-DAT database are names of cities, provinces, states or whole countries (for example, France). Geographical boundaries are necessary for the analysis, so the locations of EM-DAT are matched to spatial elements from GADM[52] using the following algorithm.

1. The reported ISO code is used to pre-select the spatial elements of GADM for the country and any attached disputed territories.
2. The reported location is prepared: replacing spatial characters (accents, numbers and punctuation); removing extra spaces; lowercase letters for all characters; synthesizing specific sentences (for example, 'Kadamjay district in Batken oblast' becoming 'Kadamjay'); correcting for any change in regional aggregation (for example, 'Haute & Basse Normandie' becoming 'Normandie'); translating any region without its variant in GADM (for example, 'Voreio Aigaio' becoming 'North Aegean').
3. Each preselected spatial element is compared with each element of the prepared reported location by applying a character matcher[53] on its names and variants.
4. Each retained spatial element is filtered using a prepared list of false positives. For instance, the location for the state of 'Ohio' triggers the identification of the county 'Ohio' in the states of Kentucky and West Virginia, which have not been reported.
5. The list of spatial elements is compared with the initial reported location, checking whether it matches correctly. If not, the issue is implemented through the prepared lists of known issues in steps 2 and 4.

Another work, the dataset GDIS, is also used to match EM-DAT locations to geographical boundaries[54]. Both approaches have been developed independently. GDIS differs in that the locations of all categories of hazards are analysed, but only up to 2018. Here, only the hazards for heatwaves were analysed, but up to 2023. Moreover, GDIS uses GADM v.3.6, whereas our work uses GADM v.4.1.

In the EM-DAT database, the dates of the heatwaves are often reported with the starting and ending days. When both days are provided, we use the average of the daily average temperature over this exact period. Other indicators may be possible[26], but the average aggregates the essential features of these heatwaves[55]. In particular, this choice is motivated by its relevance for the reported impact rather than its meteorological rarity. Heatwaves affect local populations not only through daily maximum temperatures but also through lack of cooling at night, which can be estimated using daily average temperatures. Sustaining high temperatures over time modifies the impact of a heatwave as well, justifying the use of averages over the period of the heatwave rather than its peak. The annual indicator is calculated first on each grid point, then averaged over the defined region to maximize the relevance of the indicator[1,56]. Furthermore, some events were reported without the starting and/or ending day(s). We observe that heatwaves reporting both days and lasting less than a month last on average for 8 days. Therefore, we use as an indicator for events with missing days the maximum of the 8 days running average over the reported month. In the case that several months were reported with missing starting and/or ending day(s), we lengthen the duration of the running average by 1 month for each supplementary month reported.

### Training of conditional distributions

The statistical model of attribution studies is carefully chosen to model the frequencies and intensities of extreme events[1,56]. To capture possible trends and non-stationarities, the distribution depends on the parameters driven by explanatory variables. In this study, we opt for the generalized extreme value (GEV) distribution with a linear evolution of its location as outlined in equation (1). For every year $y$ in the sample, the average temperature over the period and region of the event $T_y$ is assumed to follow a GEV distribution of location $\mu$, scale $\sigma$ and shape $\xi$, whereas the location varies with the change in global mean surface temperature smoothed over the 3 previous years (GMST).

$$T \sim \text{GEV}(\mu = \mu_0 + \mu_1 \text{GMST}, \sigma = \sigma_0, \xi = \xi_0) \tag{1}$$

Although the statistical model in equation (1) is common for the EEA of heatwaves[1,56], we have compared its performance with other potential models. Apart from this GEV model, we have tested three other distributions with linear and non-linear evolutions of the parameters: Gaussian, skew normal and generalized Pareto. Overall, our comparisons assessed through quantile–quantile plots indicate that the GEV performs the best among the four distributions, especially in terms of upper tail behaviours. We calculate the classical Bayesian information criteria (BIC) to compare their performances while reducing the risk of over-fitting[57]. We observe that for all heatwaves, a stationary GEV has the lowest BIC. We note that the linear model of equation (1) is not always the best distribution according to the BIC, although the gain in BIC from the linear model to the best solution is always marginal. More quantitatively, the improvement in BIC from the stationary GEV to the linear model represents between 88% and 100% of the improvement from the stationary model to the best solution over all heatwaves analysed, with an average of 98%. In other words, sophisticating further the statistical model would, on average, improve the performance by only 2%. This result confirms that this expression is the most appropriate for most heatwaves.

These fits are obtained by minimization of the negative log likelihood (NLL) of the training sample[58]. The first guess has its robustness improved using initial regression to approximate the coefficients[42,59]. The shape parameter is bounded between −0.4 and 0.4 (ref. 1). Moreover, the sample is weighted during minimization of the NLL, with weights equal to the inverse of the density of the GMST. This approach helps in providing equal performance over the full interval of GMST.

The choice of whether to include the event or not when estimating the statistical model has been extensively discussed, although no final consensus has been reached[1,31,60]. The results presented in this paper have been obtained by estimating the event, to prevent removing points from the observational record. To ensure numerical convergence, a minimum probability of $10^{-9}$ was set for each point of the full sample. It implies that the attributed events under factual conditions will not have return periods higher than a thousand million years, which we consider long enough.

Estimating return periods for unlikely events with relatively short observational records remains difficult[34]; thus, we append additional lines of evidence[1,56]. Conditional distributions are trained for ERA5, used as reference, but also with BEST[29] (Extended Data Fig. 1) and simulations from Climate Model Intercomparison Phase 6 (CMIP6)[30,61]. The following Earth system models (ESMs) from CMIP6 are used: ACCESS-CM2, ACCESS-ESM1-5, AWI-CM-1-1-MR, BCC-CSM2-MR, CESM2, CESM2-WACCM, CMCC-CM2-SR5, CMCC-ESM2, CanESM5, EC-Earth3, EC-Earth3-CC, EC-Earth3-Veg, EC-Earth3-Veg-LR, FGOALS-g3, GFDL-CM4, GFDL-ESM4, INM-CM4-8, INM-CM5-0, IPSL-CM6A-LR, KACE-1-0-G, KIOST-ESM, MIROC6, MPI-ESM1-2-HR, MPI-ESM1-2-LR, MRI-ESM2-0, NESM3, NorESM2-LM, NorESM2-MM and TaiESM1. For every heatwave, only the ESMs with sufficient performance are used, as described in the next section. For ERA5 and BEST, we start the time series in 1950 for adequate spatial coverage over all regions[62], and these time series finish in 2022 for BEST and 2023 for ERA5. For CMIP6, the time series is calculated over the historical (1850–2014) (ref. 63) and the SSP2-4.5 (2015–2100) (ref. 64). This scenario is chosen because its emissions are the closest to those observed over 2015-202 (ref. 36).

Only runs with the initial conditions termed r1i1p1f1 are used, as it was run by most ESMs. Only one ensemble member is used to facilitate the comparison of the parameters and probabilities from ESMs to those based on observations.

## Evaluation of the uncertainties

During the extreme event analysis, two sources of uncertainties are handled—namely, on the conditional distributions and on the handling of observations and simulations.

During the training of conditional distributions, the uncertainties on the parameters are obtained using an ensemble of 1,000 bootstrapped members, with replacements allowed during the resampling. The conditional distributions are then used to assess the probabilities and intensities of the event, under a factual climate and a counterfactual climate. The factual climate is defined as the GMST observed at the time of the event. The counterfactual climate is defined as the average of the GMST over 1850–1900.

ESMs exhibit different performance in reproducing local climates; thus, not all models may be useful for event attribution[1]. We calculate the seasonalities of ERA5 and each ESM over 1950–2020 in each grid point over the region. We then average their correlation. The most appropriate ESMs are the 10 most representative ESMs that maximize this average correlation.

Following the WWA approach, not all models are retained for further analysis[1,65]. The factual distributions at the time of the event are compared with those of ERA5. Both the scale and the shape parameters of ERA5 and the model must have their 95% confidence intervals overlapping. Otherwise, the model will be discarded. Thus, the overall selection process is to sort the ESMs by correlation with ERA5 seasonality, remove those with parameters inconsistent with ERA5 and select the 10 best ESMs in this list.

At this point, probability ratios and change in intensities are obtained for an ensemble of datasets, each with uncertainties. To synthesize over this large ensemble, equal weights are given to each bootstrap member of ERA5 and BEST, summarized into one distribution for observations. All kept ESMs are also given equal weights and summarized into one distribution for models. Finally, these distributions are averaged to deduce the median and 95% confidence intervals. We point out that synthesizing these lines of evidence could be conducted with other approaches[1], although without affecting the main messages of this work.

## Goodness of fit for the conditional distributions

Although using a non-stationary GEV with its location varying linearly with GMST is a well-established approach for EEA to statistically model extremes under global warming, this setup may not be well-suited in isolated cases[66]. To ensure that the GEV model represents the data adequately, the goodness of fit is verified for each conditional distribution used in this analysis with the method used in ref. 67.

For every heatwave, several datasets are used for analysis, from which conditional distributions are fitted. The location and scale of each of these fitted conditional distributions are used to transform the respective training sample onto a stationary $GEV(0, 1, \xi)$ with the same shape as the fitted conditional distribution. This transformed sample has observed quantiles, which are compared with the theoretical quantiles of a $GEV(0, 1, \xi)$ in a quantile–quantile plot. This quantile–quantile plot describes how well the GEV model describes the sample. The uncertainty in the GEV parameters determines a confidence band in the quantiles around the identity line, as shown in Extended Data Fig. 2.

The fraction of the sample out of the confidence band is deduced, estimated as a 95% confidence interval on the ensemble obtained from bootstrapping of the training of the conditional distribution. For each heatwave, the ensemble of conditional distributions is compared with an out-of-sample threshold at 5%. The results are shown in Extended Data Fig. 3. If the median out-of-sample fraction across the conditional

distributions is below the threshold, the goodness of fit is confirmed, and the heatwave is kept for the ensuing analysis. As shown in Extended Data Fig. 3, 217 heatwaves are retained, and nine are removed from the ensuing analysis. These nine events removed from analysis are summarized in Extended Data Table 1.

Out of the nine events, eight occurred in India, and the last one occurred in Japan. In the 217 heatwaves kept for analysis, eight events occurred in India. Besides this apparent regional clustering, no discernible traits emerge with regard to the season or length of the heatwave. More research is required to investigate why these fits do not perform as well as elsewhere for these specific events, which lies beyond the scope of this study.

## Causality using Granger causal inference

The well-established approach[1] for EEA combines observations and simulations by using non-stationary distributions. These distributions correlate the evolution of the climate indicator $T$ for the heatwave to GMST. Deducing causality, that climate change caused the event, using this correlation relies on the strong physics-based understanding[5,6,31,41,65,68–97] that increasing GMST tends to also increase regional temperatures, not only through its mean but also through the whole distribution inferred from natural variability, thus shifting the regional extremes as well. Yet, although there is a strong physical basis for this causality, we can also investigate the validity of this causality from a statistical perspective. Using Granger causal inference[98], we may assess the predictive relationship between GMST and $T$, the climate indicator of the event[4].

The common approach for Granger causality[4,98] requires that the input variables are stationary to train vector auto-regressive models[99]. This is usually verified by differentiating the variables, in other words, taking the interannual variability. This method would then assess whether the interannual variability of GMST can predict the interannual variability of $T$, thus focusing on the predictability of short-term shocks. However, the trend contains a stronger signal compared with the interannual variability. To account for long-term trends in GMST and $T$, Granger causality can be generalized using the vector error correction model (VECM)[100]. It requires the search for an adequate VECM model based on the Akaike information criterion[101] and a co-integration test, for instance, using a Johansen test[99]. Nevertheless, this method still fails to account for non-linear effects. An alternative is to use machine learning, such as Random Forest models trained to predict $T$ with GMST through their lagged effects[102]. Permutation tests are conducted to assess the performance of Random Forest models trained on permuted lagged GMST, compared with the non-permuted version[103]. Applying this method accounts for the evolution of GMST and $T$, while also accounting for non-linear effects.

By using the latter method, only three events have a median value for the test above 0.05. As shown in Extended Data Fig. 4, for 214 events out of 217, we reject the null hypothesis, concluding that the evolution of GMST is Granger-causing the evolution of $T$. The three other events are listed in Extended Data Table 2. We notice that the median value for the Granger causality remains relatively low. Using IPCC terms, it is likely (>66%) that GMST Granger-caused the evolution of $T$ for the event in the United States in 2011, whereas it is very likely (>90%) for all the others. These three events are removed from this analysis.

## Contributions from the carbon majors to global warming

The contributions of emissions of the carbon majors to global mean surface temperature are assessed with the reduced-complexity Earth system model OSCARv3.3 (refs. 104,105). The model embeds an ensemble of modules that replicate the behaviour of models of higher complexity[105]. OSCAR features the ocean and land carbon cycles with a bookkeeping module for $CO_2$ emissions from land use and land cover change, wetlands, permafrost, tropospheric and stratospheric chemistry, and global and regional climate responses to these forcers.

It accounts for the effects of greenhouse gases ($CO_2$, $CH_4$, $N_2O$ and 37 halogenated compounds), short-lived climate forcers (stratospheric water vapour, tropospheric and stratospheric ozone, primary and secondary organic aerosols, nitrates, sulfates and black carbon), surface albedo change, volcanic activity, solar radiation and contrails[37,105].

OSCAR is run over the historical period (1750–2023), following three sets of simulations: (1) The first set of simulations is driven by concentrations of greenhouse gases to ensure a match with the latest observations. (2) The second set is driven by emissions, using the compatible emissions from the first set obtained through mass balance[106,107]. This is a control run that confirms that the estimated compatible emissions lead to the observed atmospheric concentrations and is used as a reference for the following attribution runs. (3) In the third set of simulations, for each carbon major, the control run is repeated, but the $CO_2$ and $CH_4$ emissions of the major are subtracted from the compatible emissions. The difference in outcome (for example, global temperature) between the control and this simulation gives the contribution of the major. This approach is called a residual attribution method[108].

In all simulations, the radiative forcings from species or forcers that are neither $CO_2$ nor $CH_4$ (that is, forcers that are not attributed in this study) are prescribed as global time series based on the latest version of the Indicators of Global Climate Change[35]. Global time series of atmospheric concentrations for the first set of simulations come from the same source. Emissions of short-lived species (that affect the atmospheric sink of $CH_4$) are taken from the latest version of the CEDS dataset[109,110] and the updated GFED4s dataset[111] that extends the original CMIP6 emissions from biomass burning[112]. Land use and land cover change data are the same as in the latest Global Carbon Budget[36], in which we use both an updated LUH2 dataset[113] and the FAO-based dataset[114].

OSCAR runs in a probabilistic framework to represent the uncertainty in the modelling of the Earth system. This uncertainty is sampled through a Monte Carlo approach with $n = 2,400$ elements. The uncertainty in the natural processes governing the atmospheric concentration of $CO_2$ and $CH_4$ comes from the available parametrizations of OSCAR[105,115,116]. The uncertainty in the input radiative forcing follows that of the IPCC AR6 (ref. 117) and is applied uniformly to the whole time series. The uncertainty in the input land use and land cover change is sampled by running one-half of the simulations with one dataset and the other half with the other dataset. There is no uncertainty in the input emissions. Finally, the raw uncertainty range from the Monte Carlo is constrained with observational data by weighting the elements of the ensemble based on their distance to the observations in the control simulations[37,116]. As constraining values, we use decadal $CO_2$ emissions from fossil fuels and industry over 2012–2021 from the GCB[36], decadal anthropogenic $CH_4$ emissions over 2008–2017 from the AR6 (ref. 118) offset with their preindustrial value from PRIMAP third-party-based estimates[119,120], and decadal global mean surface temperature change over 2011–2020 from the AR6 (ref. 9).

### Contributions from the carbon majors to heatwaves
We assess whether the probability can be written as a sum of terms, with each term associated with contributions from anthropogenic actors or natural drivers.

We define a region in space $S$. Every year $y$, the temperature field over the region is averaged over a period $p$ of the year, then over the region $S$, resulting in the temperature $T_y$. The heatwave is characterized by the exceedance of the heatwave level $u$ by $T_y$, with $u$ a real-valued scalar. $T_y$ represents a real-valued continuous random variable (Borel $\sigma$-algebra on the reals). Given the heatwave level $u$, the target probability is a survival function $P(T_y > u)$.

We assume that the probability of the heatwave is conditional on $GMT_y$ and that it follows the statistical model introduced in equation (1) and represented in equation (2). Every year, the temperature over the region and the period $T_y$ is sampled from a non-stationary GEV

distribution[27]. The parameters of this GEV distribution are the location $\mu$, the scale $\sigma$ and the shape $\xi$. The location varies linearly with a covariate, the change in $GMST_y$ at the corresponding year.

$$P(T_y > u | GMST_y) = 1 - GEV(u | \mu = \mu_0 + \mu_1 GMT, \sigma = \sigma_0, \xi = \xi_0) \qquad (2)$$

With the analytical expression for the cumulative distribution function of the GEV that follows equation (3):

$$GEV(u | 0, 1, \xi) = \begin{cases} \exp(-(1 + \xi u)^{-1/\xi}) \text{ for } \xi \neq 0 \text{ and } 1 + \xi u > 0 \\ \exp(-\exp(-u)) \text{ for } \xi = 0 \end{cases} \qquad (3)$$

This well-established statistical model is widely used for EEA[1,56] and has already been used extensively for heatwaves. We acknowledge that a more sophisticated model with additional covariates may further improve the performance[121,122]. However, this statistical model has been shown to have good performance for heatwaves in general[13], and additional covariates can prevent the use of climate models as additional lines of evidence. The former section provides additional grounds for the choice of this model.

The causal theory applied to climate change justifies the decomposition of probabilities in a Gaussian case[123,124]. Given a statistical model built on a non-stationary Gaussian distribution linearly driven by GMST, if GMST can be split into a sum of contributions, then the probabilities can be approximated as a sum of their associated contributions[123,124]. However, the statistical model presented in equation (2) uses a GEV instead of a Gaussian. Even by attempting to write the decomposition using Bayes's theorem and the inclusion–exclusion principle, the exact analytical form of each term remains challenging. This is mostly because the differences in probability when removing a contribution to GMST depend on the initial value of GMST. In other words, the non-linearity and the high number of terms lead to a solution that cannot be computed exactly.

Instead, we propose to approximate the solution and to investigate the quality of this approximation. The usual approach to calculate contributions to climate change is to run the statistical model with all contributors, then to run it again without one contributor, the difference corresponding to the contributor. This approach is thereafter called All-But-One (ABO). Thus, an emitter $e$ with a contribution to global warming $GMST_{y,e}$ would contribute to the probability of the event using this approach.

$$P_{y,e}^{ABO} = P(T_y > u | GMST_y) - P(T_y > u | GMST_y - GMST_{y,e}) \qquad (4)$$

To account for non-linear effects in the decomposition of probabilities, this approach is complemented with a second approach that calculates the difference in $GMST_y$ introduced by adding only the emitter (Add-One-to-None, AON). According to this approach, the emitter $e$ would contribute to the probability of the heatwave as follows:

$$P_{y,e}^{AON} = P\left( T_y > u | GMST_{y,e} + GMST_y - \sum_e GMST_{y,e} \right) \\ - P\left( T_y > u | GMST_y - \sum_e GMST_{y,e} \right) \qquad (5)$$

The approach based on the removal of a single entity (ABO) estimates the contribution of a state perturbed by all the other contributors. The approach based on the addition of a single entity (AON) evaluates the contribution in an unperturbed state, without the other contributors interfering. Given the non-linearity of the system, we expect the physical contribution to be between the two values. We choose to calculate both approaches and average them. This approach, calculated using equation (6), is called the combined ABU & AON (Extended Data Fig. 5).

$$P_{y,e} = \frac{P_{y,e}^{ABO} + P_{y,e}^{AON}}{2} \tag{6}$$

For each event, the probability is calculated for all datasets for the region and averaged over the datasets. Its 95% confidence interval is calculated using bootstrapping. The total probability of the event is decomposed into contributions of each carbon major, other climate forcers and preindustrial probability. After decomposition, these terms are summed up for comparison with the total probability. The 95% confidence interval is shown for all events, and only one event (Cyprus, May–September 2022) does not reproduce the total probability. This event has been removed from the analysis of extreme events. As shown in Extended Data Fig. 5, the average of ABO and AON provides the best estimate, because it accounts for non-linear effects.

In EEA, probabilities are often communicated using probability ratios, quantifying how many times climate change has made the event more likely. It is calculated using the probability of the event in a preindustrial climate, thus with a GMST averaged over 1850–1900:

$$PR = \frac{P(T_y > u | GMST_y)}{P(T_y > u | GMST_{1850-1900})} \tag{7}$$

Because the contribution of the emitter $e$ to the probability of the heatwave $P_{y,e}$ is a perturbation, the emitter multiplies the probability of the heatwave as in equation (8):

$$PR_e = 1 + \frac{P_{y,e}}{P(T_y > u | GMST_{1850-1900})} \tag{8}$$

## Discussing an alternative decomposition approach

Alternatively to the approach based on GMST, a basic approach would be to assess the contributions directly with the emissions. The fraction in the cumulative emissions at the time of the event would represent the share of responsibility of the carbon major in the causes of the event. This fraction can be used for the change in intensity and the change in probability of the event. This approach can be compared with the principle applied for the attributional life cycle assessments, taking the Earth system as a whole and using the shares in its inputs to trace the perturbation[125,126], whereas the approach based on GMST traces the effects of the carbon majors through the Earth system. Therefore, GMST is more similar to the principle of the consequential life cycle assessment. However, the approach based on cumulative emissions has several drawbacks.

First, the carbon majors fuel climate change with $CO_2$ and other compounds, such as $CH_4$. As an approximation, it would still be possible to aggregate these compounds using a global warming potential for fossil $CH_4$.

Then, the carbon cycle partially absorbs the emitted carbon over time. Thus, two companies with the same cumulated emissions may not share the same responsibility, if one has older emissions, thus with a lower contribution to the atmospheric concentration of $CO_2$. Still, these old emissions contributed to warming up the Earth system and saturating the carbon sinks.

Finally, the attribution analysis may not respond linearly to changes in GMST. For our study, heatwaves are represented with a GEV with the location varying linearly with GMST. According to the Transient Climate Response to Emissions (TCRE), the GMST varies almost linearly with cumulative emissions. Thus, the approach based on cumulative emissions would lead to similar results to ours. However, for events for which the distributions do not vary linearly with GMST, as it may for extreme precipitations[1,56], non-linearities would be introduced.

To conclude, the approach based on cumulative emissions is an approximation that relies on the linearity of the Earth system. However, this system is not entirely linear, and the TCRE is known as an approximation with its limits[127]. Under the assumptions that the linearity of the system would be respected, this simple approach would then lead to similar results as those based on the approach used in this work.

## Data availability

The data that support the findings in this study are available through the following references: disaster database EM-DAT (https://public.emdat.be/), geographical boundaries database GADM (https://gadm.org/download_world.html), Carbon Majors database (https://carbon-majors.org/Downloads), ERA5 (https://cds.climate.copernicus.eu/datasets/reanalysis-era5-single-levels), BEST (https://berkeleyearth.org/data/), and CMIP6 on the Earth System Grid Federation data nodes (http://esgf-node.llnl.gov/search/cmip6/). Detailed data for the search query are as follows: Experiment ID (historical, ssp245), Variant Label (r1i1p1f1), Frequency (day) and Variable ID (tas). The outputs of this study are provided in the Supplementary Information.

## Code availability

All the codes that support the findings in this study are available at Zenodo[128] (https://doi.org/10.5281/zenodo.15569401). Moreover, the code for OSCARv3.3 can be accessed at Zenodo[129] (https://doi.org/10.5281/zenodo.10548477).

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

**Acknowledgements** This work was supported by funding from the Horizon 2020 and Horizon Europe research and innovation programmes of the European Union (grant agreement nos. 101003687 (PROVIDE), 101081369 (SPARCCLE) and 101003469 (XAIDA)). T.G. also acknowledges support from the Horizon Europe research and innovation programme of the European Union (grant agreement no. 101056939 (RESCUE)). R.H. acknowledges funding from the Rockefeller Brothers Fund. We thank L. Pierini for verifying the equations for the decomposition of contributions. W.T. acknowledges funding from the European Research Council under the Horizon Framework research and innovation programme of the European Union (grant agreement no. 101124572 and European Research Council consolidator grant 'LACRIMA'). Views and opinions expressed are, however, those of the authors only and do not necessarily reflect those of the European Union or the European Research Council Executive Agency; neither the European Union nor the granting authority can be held responsible for them. We thank the referees for their careful review of the paper.

**Author contributions** Y.Q., L.G., D.L.S., T.G. and S.I.S. conceptualized the study. Y.Q. developed the software and carried out the analysis. R.H. provided guidance on the Carbon Majors database. T.G. conducted the analysis with OSCAR. P.N. revised the statistical decomposition. Y.Q. drafted the text, and Y.Q., L.G., D.L.S., T.G., R.H., C.H., Q.L., S.N., P.N., W.T., C.-F.S. and S.I.S. contributed to interpreting the results and refining the text.

**Funding** Open access funding provided by Swiss Federal Institute of Technology Zurich.

**Competing interests** The authors declare no competing interests.

**Additional information**
**Correspondence and requests for materials** should be addressed to Yann Quilcaille.

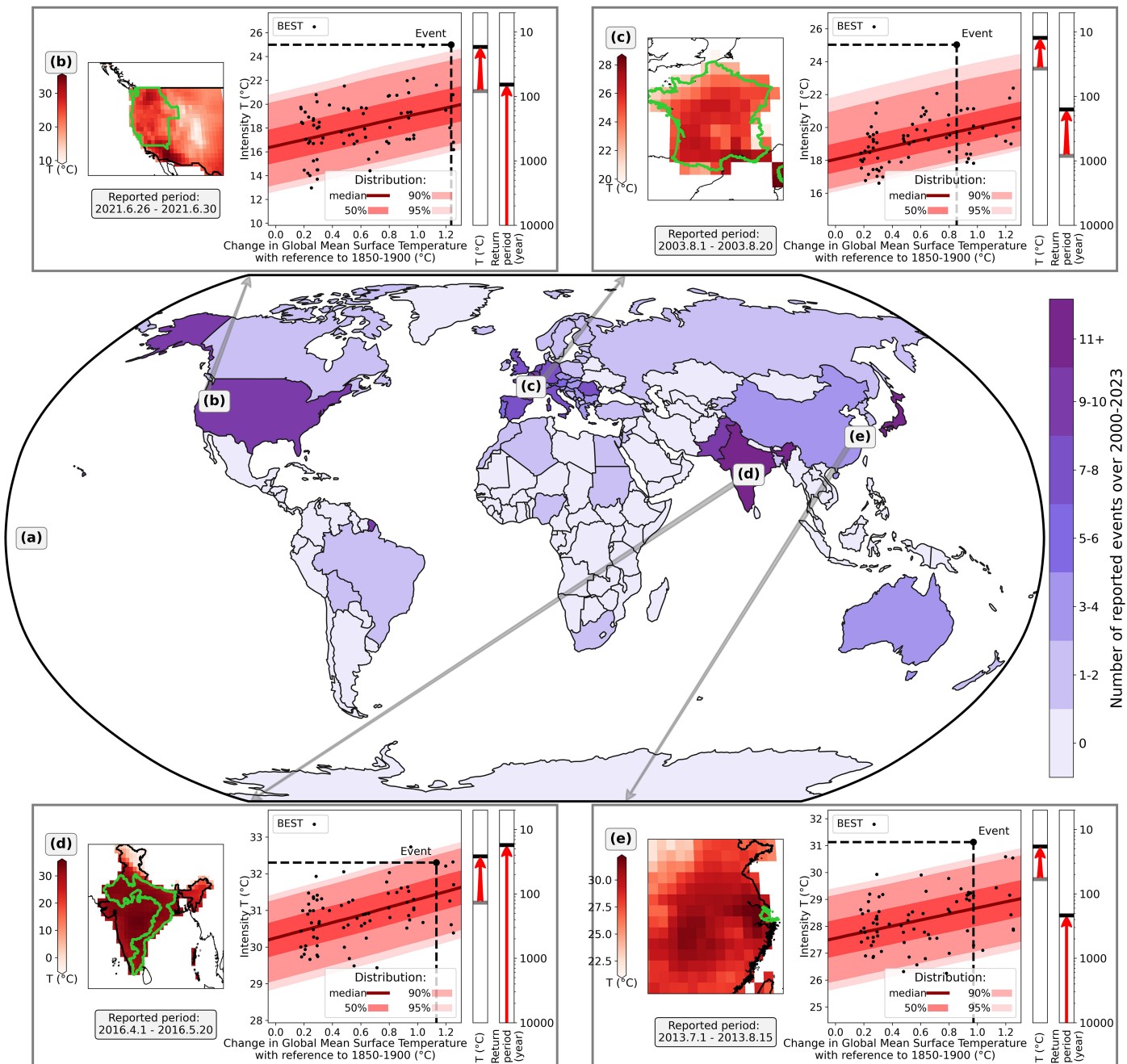

**Extended Data Fig. 1 | Besides ERA5, the contributions of climate change to each event are also assessed based on other datasets, such as BEST[29].** **a**, Number of heatwaves reported per country in EM-DAT over 2000–2023. An extreme event attribution is performed for each of them, as illustrated for four examples with BEST[29] data: **b**, the 2021 Pacific Northwest heat dome, **c**, the 2003 heatwave in France, **d**, the 2022 Indian heatwave, and **e**, the 2013 eastern China heatwave. For each example, the average temperatures during the event are mapped, with the outlines of the reported region (lime green contours).

In addition, the intensity (average temperature T (°C) during the period and region of the event) and change in Global Mean Surface Temperature (GMST, °C) are represented over 1950–2023 (black dots), with their conditional distribution represented through the median (red line) and ranges of the distribution (red shading). Finally, the change in intensity and change in return period (year) compared to the pre-industrial reference period are shown for each example. Uncertainties inferred using bootstrapping are not shown here for the sake of clarity.

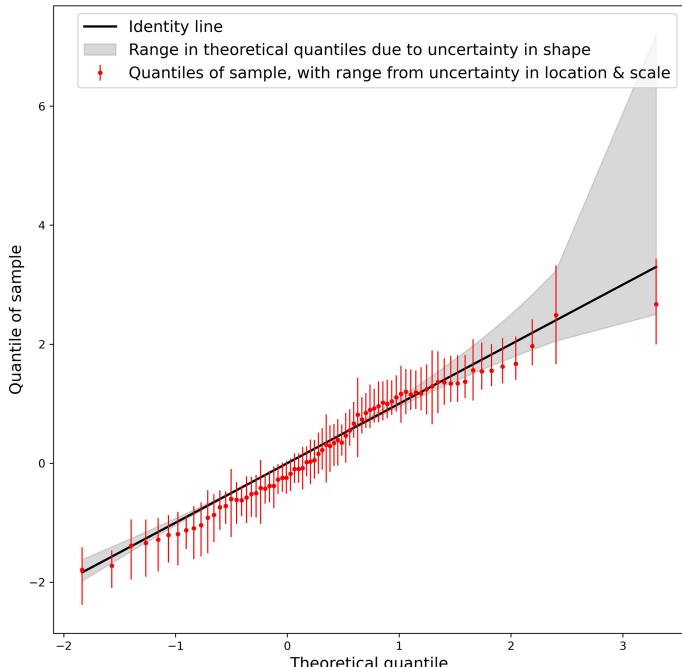

**Extended Data Fig. 2 | Illustration of the goodness-of-fit with the quantile-quantile plot for the heatwave Pacific North West 2021 using ERA5[28].** For a perfect fit, the quantiles of the sample (red dots) would be distributed over the identity line (black line). The uncertainties on the shape parameter provide a confidence band for the theoretical quantiles (grey shaded area), while the uncertainties on the location and scale provide a confidence band for the quantiles of the sample (red lines).

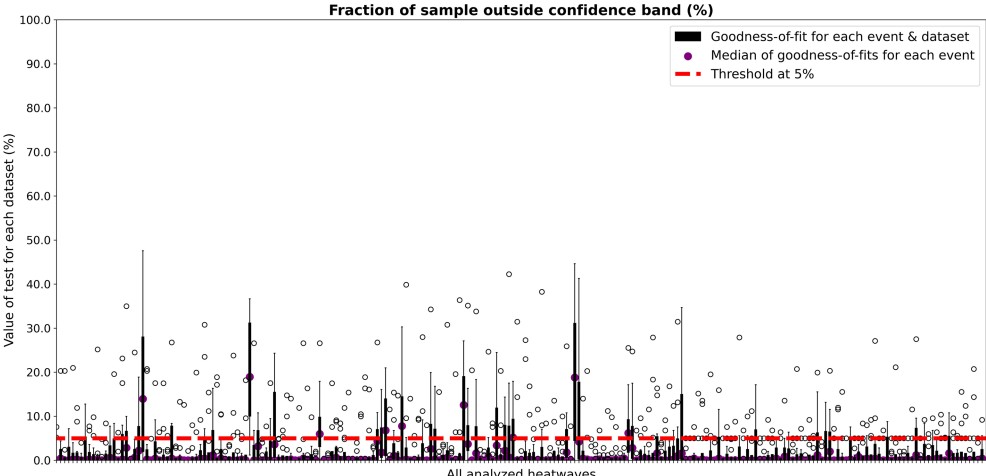

**Extended Data Fig. 3 | Out of the 226 analysed heatwaves, 217 events show sufficient goodness-of-fit.** For each of the heatwaves and datasets used for analysis, the fraction of sample outside confidence band is calculated (black circles). An event is rejected if the sample has a fraction higher than 5% out of the confidence band (red dashed line). For each heatwave, results are summarized over datasets in boxplots (black box) and the median (purple dot).

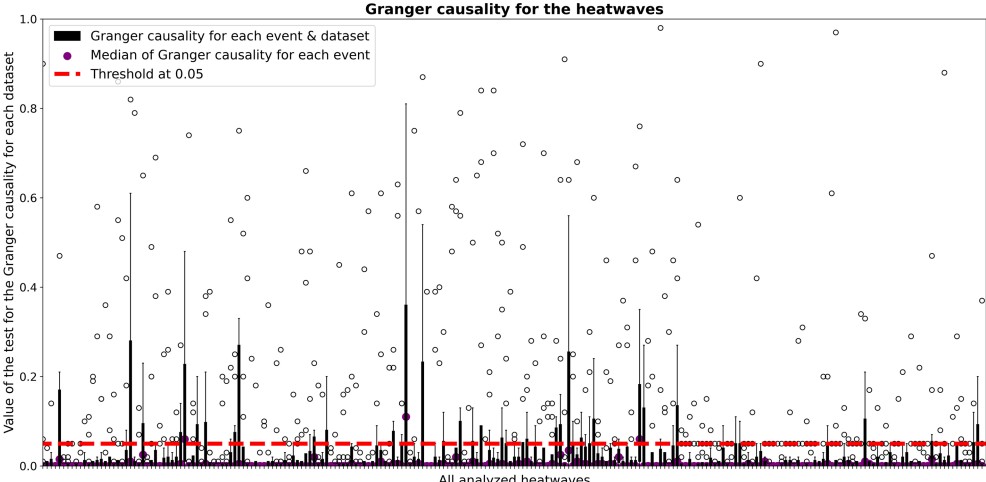

**Extended Data Fig. 4 | Out of the 217 analysed heatwaves, 214 events demonstrate a significant causality.** For each of the heatwaves and datasets used for analysis, the non-linear Granger causal inference is tested through comparisons of performance of Random Forest models (black circles). The hypothesis of non-causality is rejected if the p-value is lower than 0.05 (red dashed line). For each heatwave, results are summarized over datasets in boxplots (black box) and the median (purple dot).

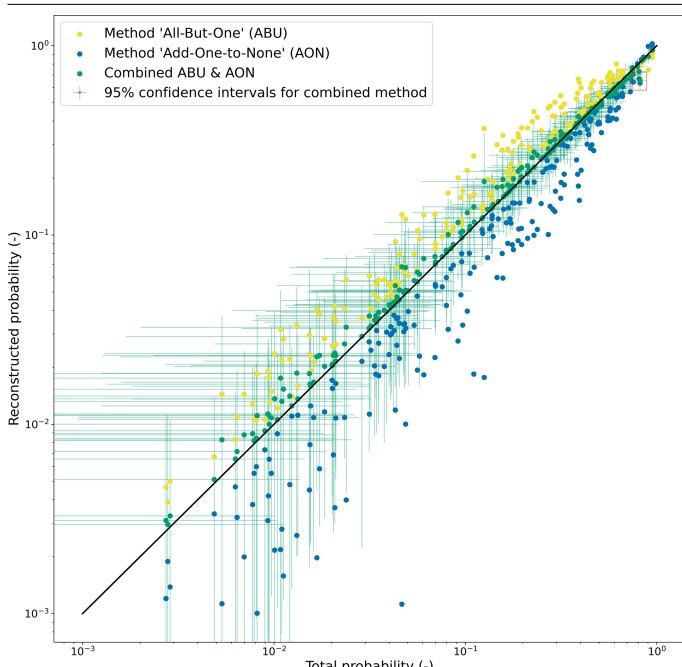

**Extended Data Fig. 5 | To decompose probabilities, the method extends well-established approaches for higher performance, shown sufficient for 213 heatwaves out of 214 events.** Each heatwave attributed in this analysis is shown here (coloured dots). The total probability of each event is the result from the attribution analysis. The reconstructed probability results from the decomposition of the probabilities using the three approaches (ABU in yellow, AON in blue, combined in green), then all contributions are summed up to facilitate the comparison. The event in the red box in the upper right corner is the only event for which the decomposition fails. The 95% confidence interval is shown for the combined ABU & AON (green lines) only for the sake of clarity.

**Extended Data Table 1 | Details of the nine heatwaves removed from analysis due to lack of goodness-of-fit of the conditional extreme value distributions compared to selected other heatwaves**

| Disaster in EM-DAT | Country & Region | Period | Goodness-of-fit |
|---|---|---|---|
| 2005-0323-IND | India: Uttar Pradesh, Bihar, West Bengal, Orissa, Maharashtra, Andhra Pradesh provinces | 2005.6 - 2005.6 | 18.9% |
| 2019-0217-IND | India: Patna, Gaya, Bhagalpur cities; Aurangabad, Nawada districts (Bihar) | 2019.6 - 2019.6 | 18.8% |
| 2003-0250-IND | India: Andhra Pradesh, Orissa, Tamil Nadu, Vidarbha (=part of Maharashtra), Chhattisgarh, Jharkland, Uttar Pradesh, Rajasthan, Madhya Pradesh, Bihar provinces | 2003.5.14 - 2003.6.6 | 13.9% |
| 2015-0189-IND | India: Delhi, Andhra Pradesh, Orissa provinces | 2015.5.20 - 2015.5.31 | 12.5% |
| 2010-0206-IND | India: Delhi, Haryana, Madhya Pradesh, Orissa, Rajasthan, Uttar Pradesh provinces | 2010.3 - 2010.5 | 7.8% |
| 2009-0175-IND | India: Orissa, West Bengal, Bihar, Uttar Pradesh, Jharkhand, Andhra Pradesh provinces | 2009.4.14 - 2009.6.26 | 6.8% |
| 2019-0463-JPN | Japan | 2019.7.29 - 2019.8.8 | 6.2% |
| 2007-0207-IND | India: Uttar Pradesh, Rajasthan, Punjab, Delhi provinces | 2007.4 - 2007.6 | 6.0% |
| 2017-0144-IND | India: Andhra Pradesh, Telangana, Maharashtra, Odisha (Orissa) | 2017.4.12 - 2017.6.15 | 5.1% |
| 2019-0223-IND | India: Karnataka, Maharashtra, Madhya Pradesh, Rajasthan, Uttar Pradesh states | 2019.6 - 2019.6 | 4.2% |
| 2006-0237-IND | India: Uttar Pradesh, Orissa, Punjab, Delhi provinces | 2006.5 - 2006.5 | 3.6% |
| 2013-0208-IND | India: Adilabad, East Godavari, Guntur, Karimnagar, Khammam, Krishna, Nalgonda, Nizamabad, Prakasam, Vishakhapatnam, Warangal, West Godavari districts (Andhra Pradesh province) | 2013.4.1 - 2013.5.30 | 2.6% |
| 2016-0133-IND | India: Andhra Pradesh, Rajasthan, Orissa, Bihar, Jharkhand, Tamil Nadu, Uttar Pradesh provinces | 2016.4.1 - 2016.5.20 | 2.2% |
| 2002-0290-IND | India: Madhya Pradesh, Andhra Pradesh, Uttar Pradesh, Delhi, Rajasthan, Punjab, Haryana, Orissa, West Bengal provinces | 2002.5.10 - 2002.5.22 | 0.2% |
| 2022-0248-IND | India: Maharashtra State, West Rajasthan | 2022.3 - 2022.4 | 0.0% |
| 2023-0234-IND | India: Maharashtra state | 2023.4 - 2023.4 | 0.0% |
| 2023-0328-IND | India: Uttar Pradesh, Bihar, Tripura, Odisha | 2023.4 - 2023.6 | 0.0% |

The location, period and value of the median goodness-of-fit across datasets are listed for each heatwave with insufficient performance. Additionally, the information on other heatwaves occurring over India but kept for analysis are also provided as a point of reference.

**Extended Data Table 2 | Details of the three heatwaves removed from analysis due to lack of significant Granger causality**

| Disaster in EM-DAT | Country & Region | Period | Granger causality |
|---|---|---|---|
| 2011-0264-USA | USA: New York, Washington, Texas, Ohio, Maine, Iowa, Illinois, Minnesota, Nebraska, Wisconsin, South Dakota provinces | 2011.7.19 - 2011.7.22 | 0.11 |
| 2003-0391-NLD | The Netherlands: Drenthe, Flevoland, Friesland, Gelderland, Groninge, Limburg, Noord-brabant, Noord-holland,Overijssel, Utrecht, Zeeland, Zuit-holland provinces | 2003.7.31 - 2005.8.13 | 0.06 |
| 2020-0530-BEL | Belgium | 2020.8.5 - 2020.8.8 | 0.06 |

The location, period and value of the median Granger causality across datasets are listed for each heatwave with insufficient performance.