## [Peer Review file · Nature]

Systematic attribution of heatwaves to the emissions of carbon majors

Corresponding Author: Dr Yann Quilcaille

Version 0:

Reviewer comments:

Referee #1

(Remarks to the Author)

The article conducts a systematic event attribution of 187 historical heatwaves, and attributes the effects to individual carbon majors. The novelty lies in the fact that this is such a large systematic study and the results are therefore comprehensive and relevant to a wide range of readers, the attribution to carbon majors is also an interesting development. As such I think that this paper is worthy of being published in this journal.

My main concern is that at times it is hard to determine what is new here, I think this would be the case for the many non-specialist readers of this journal. For example which of the methods being employed are themselves novel and which are identical or very similar to those already being used in other studies. While many of these studies are referenced and an expert or interested reader could discover this themselves I think it would be helpful to include more phrases such as “following ...” or “similar to ..” to make it very obvious when an existing method is being used.

Reading the section beginning L44 might also give the impression that this is the first study to attempt to attribute extreme events to individual sources – so care needs to be taken to ensure that the reader realises that there are already several studies linking extreme events to countries (e.g. Otto et al 2017) and individuals (e.g. Lott et al 2021).

The article should also be clearer if this attribution is to anthropogenic effects or climate change in general as it seems to be described both ways e.g. L24 vs L113.

In several places you describe how climate change has increased the risk and amplitude by a certain amount but do not specify from when (I assume this is 1850-1900) but it would be good to be more explicit, especially in the abstract.

The values for attributable risk seem quite high. For the example presented for the Pacific NW heatwave, the previously calculated value of 150 times on L96, although consistent with your uncertainty, is much smaller than your median (over 10000). Can you comment on this and perhaps give an example of such large numbers being calculated in previous literature?

L308 – could you give a bit more detail on why this is the best method for model selection – or give a reference where this is also done – as its not immediately clear to me why a model with good seasonality would be best at modelling the frequency of extreme events.

Minor comments:

Figure 1 – could you make the outline of the events a bit clearer.

L277 – perhaps change “former” to “previous” as I think this would be clearer.

L284 – please explain what NLL means – i.e. negative log likelihood.

Could you consider changing GMT to GMST – as I think this is more commonly used – this is also the abbreviation that is

used for this metric by the IPCC

(Remarks on code availability)

Unfortunately I have run out of time to try to get this code to actually work (its quite complicated so this would take longer than I have).

However I have looked through it and can confirm that there is a README file which looks much more helpful than most I've seen and the code itself is well structured and commented so it all looks good to me.

Referee #2

(Remarks to the Author)

General comment:

The authors present a systematic approach to attribute altered heat wave risks to the contribution of individual polluters, commonly referred to as carbon majors. They pick the most reliable time interval (2000-2022) and estimate changes in intensity and frequency of occurrence for 187 heat waves that occurred during that period of time according to the most widely used international disaster database (EM-DAT). In doing so, they clearly state that the database is far from perfect and very uneven in its spatial representation of extreme heat events. The advantage of that approach is that they essentially circumvent the problem of event definition. On the flip side, the 187 events the database are a function their societal impacts and thus a biased view as to how relevant they are. An issue, which is hard to overcome which is why i concur with their argument that using EM-DAT is a defensible approach for the purpose of this analysis.

Using observational data, reanalysis and Earth system models, the authors attribute each heatwave individually, i.e. they quantify how anthropogenic climate change has affected each heatwave in terms of changing intensity and likelihood of it occurring. Based on the widely used GEV distribution, they present 4 test cases as example to showcase the viability of the method. The method itself is also neatly done in that they tackle the problem of short observational records using climate model simulations from CMIP6. This way, they are able to quantify the extremely low odds of observed events occurring under counterfactual conditions. Not only that, they test the respective model ensemble for its fitness for purpose much in same way World Weather Attribution does (clear thresholds for GEVs shape and scale parameter that have to be met at the respective grid point level when compared with reanalysis). Common uncertainty sources are analysed using bootstrapping.

Splitting the 2000-2022 period into 3 time intervals (2000-2009, 2010-2019, 2020-2022), the authors are able to demonstrate how the change in risk and intensity has drastically increased over time. While at varying magnitude, they found that all 187 heat waves have become more intense and are now occurring with higher probability than without human-induced climate change. Unsurprisingly, the change in intensity is higher on median average (2.0°C for 2020-2022) than the global temperature increase relative to 1850-1900 itself (1.25°C in 2022), very much in line with many previous attribution studies.

In the final step, they attribute the heatwaves to the carbon majors individual contributions. That's where the paper really covers new ground. First they calculate the contributions from the carbon majors to global warming applying a reduced-complexity Earth system model (OSCAR). And second, they decompose the entities contributions to the heatwaves. The first step is straight-forward in that entities emissions are converted into a radiative forcing contribution with an associated fractional change in global mean temperature. Again, the analysis is accompanied by a thorough uncertainty analysis for the uncertainty in the modelling of the Earth system using Monte Carlo sampling. The second step is where things get interesting. Applying Bayesian statistics (Bayes' theorem) and an Inclusion-Exclusion principle, they are able to deduce the probability ratio of the carbon majors to each heatwave as a function of their previously estimated contribution to the global temperature. The approach even allows to account for potential non-linear effects in heat wave frequency with warming.

The authors find that 0.68°C of the 1.25°C in 2022 are caused by 122 carbon majors, with 0.32°C attributable to the Top 6 entities. While staggering as such, previous studies came to similar conclusions. The important aspect is the contribution of individual majors to an increased intensity of individual heatwaves. On average, the contribution is between zero and 0.33°C to all (individual) heatwaves. It does not sound much, but knowing that one company may have made a specific heat wave warmer by up to 0.33°C is a rather crucial finding. Not only from a scientific, but also decision making and (climate) communications perspective, given such results are much more tangible for lay people and experts alike. Sure, most events and most companies have much smaller footprints, but the fact that these contributions can be assigned to individual polluters is - in my view - a milestone. The attributed change in risk of occurrence is equally interesting, though less intuitive for the target audience.

Summary:

As you can certainly tell from my rather enthusiastic general review comment (quibbles thereafter), I wholeheartedly recommend the paper for publication in Nature given its thoughtful state-of-the-art approach and analysis, as well as its careful consideration of the available observational data and modelling tools. In other words, quality of the data as well as the representation is excellent. Context and clarity is provided (with a few exceptions raised below). The authors convincingly demonstrate that the results are robust within the bounds of widely accepted uncertainty analysis standards. As a note of caution, I will say that Bayesian statistics is not my strong suit (to put it mildly), hence I do put a fair amount of trust in them applying the respective methods (Bayes' theorem and the Inclusion-Exclusion principle) with best intentions and due diligence. Their discussion of the caveats and the overall impression of the presented manuscript makes me very

confident that the analysis stands the test of better stats experts than I am. That all said, I do have a few questions that I would like the authors to be addressed before possible publication.

Minor comments and quibbles:

Lines 202-204 (and 231-233): "The heatwaves over 2000-2022 had their intensity increased on average by 1.59 °C due to climate change, of which 0.40 °C is due to the seven top carbon majors, and 0.41 °C is due to the 115 others."

—> It took me several times reading this, until I figured where it's probably coming from. Am I correct to assume that it is the average of the 2000-2009, 2010-2019, 2020-2022 intensity increase (of all 187 heatwaves) presented in the section above? What leaves me still puzzled is the fact that you refer to the decadal results (1.3°C, 1.6°C, 2.0°C, resp) as median average, where here it's just average ... so what is it? Also, I suppose that the 0.81°C for all 122 carbon majors is the result based on the OSCAR model? Please clarify and try to improve the manuscript at this point in order to make it easier for other to follow.

The discussion section could be made a lot more exciting. To be honest, it felt a bit flat after reading through an otherwise great paper. It contains a lot of (partially literal) repetition of what was just said in the previous paragraphs. Feels a bit of a waste given the strict space limitations. For example, the legal repercussions of the findings could be discussed, but I could also do with an expansion from a caveats perspective. One concrete aspect in terms of caveats is the impact of the counterbalancing (i.e. cooling) aerosol effect, that may enhance the carbon majors warming contribution even further. It's briefly mentioned in lines 242-247, but perhaps there's room to assess things especially in terms of qualitative aerosol impact estimates.

Line 211: What are CNX resources? Acronym not referenced.

Figure 1: Would it make sense to have the same return period interval on the bars on the rhs in each of the 4 station panels? It's obvious why it's different for temperature, but not why it's different for the return period labelling. Also, is that same figure available for BEST (to be put in the supplementary material)?

(Remarks on code availability)

Two of my colleagues and I did take a look at the code and found it very well explained and structured. Given that a massive amount of CMIP6 data need downloading before the scripts can be run and tested, we did not have the time to do that. So I do not know whether this counts as having reviewed the code, but I reckon it might, so I put 'yes'.

Referee #3

(Remarks to the Author)

I have been asked, if the findings of this paper, are technically robust, whether they would have important implications for legal and/or regulatory actions.

This paper makes an important contribution to the growing literature on climate attribution. There are now approximately 25 lawsuits pending in the courts in the United States, and a small number in other countries, where climate attribution would be relevant. The suits typically seek money damages from the fossil fuel majors for the costs of adaptation to climate change. A few of these cases have been dismissed. The others were held up for years by various jurisdictional questions. None of them has gotten to trial or even to discovery on the merits, and thus to issue of attribution has not yet been litigated.

One of the pending cases was brought by the City and County of Honolulu against several fossil fuel majors. The Hawaii Supreme Court ruled for the plaintiffs. The defendants have asked the U.S. Supreme Court to take the case. The Supreme Court has asked the Solicitor General of the U.S. for her views on whether they should take it; she has not yet responded. This inquiry shows that the Supreme Court may be interested in taking the case. If they do take the case, it is entirely possible that they will issue a ruling that has the effect of wiping out all 25 of these cases. If that happens, this paper may be of little legal importance in the U.S. However, if the Supreme Court refuses to take the case, or if it takes the case and affirms the holding of the Hawaii Supreme Court, then this paper would be of great significance. I expect that it would be cited by the plaintiffs in most or all of these cases. The defendants will then try very hard to discredit it. Thus thorough technical peer review is especially important.

The paper is also relevant to international negotiations on the issue of loss and damage — how much money should the developed countries pay the developing countries to compensate them for their losses from climate change. There may be no judicial remedy to seek loss and damage, and ultimately the developed countries (or some of them) may refuse to pay significant amounts, but this paper will certainly be of interest to the negotiators from the climate-vulnerable countries.

(Remarks on code availability)

Referee #4

(Remarks to the Author)

See attached pdf report.

(Remarks on code availability)

Version 1:

Reviewer comments:

Referee #1

(Remarks to the Author)

The article conducts a systematic event attribution of 187 historical heatwaves, and attributes the effects to individual carbon majors. This is my second review of the article and I thank that authors for considering all my comments and find the current version much improved, particularly regarding where this present paper relates to existing literature.

The major remaining concern I have regards the event definition. What exactly are the black dots on figure 1? From reading the text and caption my understanding is that they are Tmax averaged over the region and period of the event – with one value per year. I find this quite a strange choice and do not understand why have you chosen this definition rather than say annual maximum values averaged over the length of the event which I think are the more typical and robust approaches? Indeed, the studies that you compare to for the Pacific NW event use annual maximum Tmax (TXx) and a 5-year block maximum of annual maximum seven-day running mean air temperature.

Could you clarify and justify your choice of approach, particularly given that you are using a GEV fit which as I understand it is only applicable to fit to distributions of extreme values. The choice of how to define an event is a key part of event attribution so I think this point needs more detail in the main text.

Minor point

Lines 97-101: “Using the sole” would be clearer as “using only”. The sentence following this seems a bit repetitive in terms of phrasing and it would be good to rewrite this.

Line 139. What does “best” mean in this context?

(Remarks on code availability)

I am satisfied that the code is available and useable.

Referee #2

(Remarks to the Author)

I finally managed to go through all the referee comments and responses, with apologies for the delay in doing so.

I am impressed by the rigour and detailedness of the responses by the author team. Also, having read the critical review of Referee #4, I can absolutely understand why the authors challenged the decision to reject the manuscript. I have to say, most of their (the referee's) points were either based in misunderstandings or a general unwillingness to accept standard methods in EEA. In particular some of the comments made on the event definition by referee #4 border on straw man arguments. I admire the calm and considered response by the authors and the massive amount of extra work they invested to address their critique exhaustively.

The other two referee comments were addressed with equal care. Needless to say, my initial points of criticism have all been addressed and are now appropriately amended in the manuscript. Very well done! Therefore, I happily accept the paper as is and wholeheartedly recommend publication in Nature.

(Remarks on code availability)

Referee #5

(Remarks to the Author)

Dear authors,

I have been asked to assess your replies to the now-absent referee #4 (R4). Upon reading the comments from the previous R4 and enough of the manuscript to grasp the core of your motivation, methods, and results, my overall feeling is that R4 was overly (and unnecessarily) critical of your manuscript and statistical methodology. I would agree with R4 that the methods used here are not necessarily state-of-the-art in the statistics literature; however, I also agree with the authors that some balance between sophistication and practicality/reproducibility is needed for large-scale studies such as this one. Ultimately, I think you have graciously revised your manuscript in light of R4 feedback while respectfully standing your ground in areas where R4 was (in my opinion) unreasonable.

That said, I do have a few minor concerns that I think should still be addressed:

1. R4 Comment 2.1(a)

Here, R4 was asking about the appropriateness of using GEV theory for analyzing block maxima from finite data while extrapolating very far into the upper tail (in some cases, to >10,000-year events). They correctly raise the point that the measurements from which you take a block maximum (daily T within a year) are definitively neither independent nor identically distributed. R4 then asks you to conduct goodness-of-fit tests to ensure appropriateness of the GEV. In your revision, the Supplementary Information now states:

“Besides this GEV model, we have also tested three other distributions with linear and non-linear evolutions of the parameters: Gaussian, skew normal and generalized Pareto. Overall, our comparisons assessed through quantile-quantile plots clearly indicate that the GEV performs the best among the four distributions, especially in terms of upper tail behaviours.”

While it is nice to know that the GEV provides the *best* fit from among the other candidates, this does not imply that the GEV does indeed provide a *good* or *appropriate* fit to the data (while the GEV may be best, it could still provide a poor, albeit relatively good, fit).

Related, in the Supplementary Materials you also state:

“The choice of including or not the event when estimating the statistical model has been extensively discussed, although no final consensus was reached^{7,16,17}. The results presented in this paper have been obtained by estimating with the event, to prevent removing points from the observational record. To ensure numerical convergence, a minimum probability of $10e-9$ was set for each point of the full sample. It implies that the attributed events under factual conditions will not have return periods higher than a billion years, which we consider long enough.”

I concur that there is no final consensus on whether the event of interest should be considered in- or out-of-sample for the analysis. However, whether or not it is included can have major implications for the associated return periods and probabilities. As you probably know, this issue reared its ugly head in the PNW heatwave. Philip et al. (2022) directly found that treating the 2021 temperature as out-of-sample yielded an infinite return interval for the event; hence, they elected to include the event in their analysis to yield a finite return interval (and, therefore, very different return intervals with and without the event of interest). Bercos-Hickey et al. (2022) directly looked at goodness-of-fit and found that for many of the stations considered the GEV distribution did *not* provide a good fit to the data when the 2021 measurement was included, but when it was excluded, the GEV distribution was acceptable.

In my opinion, you need to consider this goodness-of-fit question more formally, particularly since (1) you include the heatwave events as in-sample and (2) you use the GEV fits to extrapolate far into the upper right tail. I'd suggest something along the lines of what was done in Bercos-Hickey et al. (2022); also, Risser et al. (2025) <https://doi.org/10.1016/j.wace.2025.100743> propose methods for conducting these checks (see their Section 3.1 and also Section 2.2.5 of the supporting information).

2. R4 Comment 1.5

I think the reviewer is asking about something else here, but I concur with the overall sentiment regarding the importance of clearly stated causal assumptions. I find your approach for isolating the carbon majors (averaging the ABO and AON fits) both novel and interesting, and Figure S2 is compelling. However, particularly for the observational data sources, what you are doing (regressing one time series on another with GEV errors) is still subject to problems associated with hidden covariates (i.e., you find correlation and not causation). In some ways, your approach could be framed as Granger causality (Granger, 1969 <https://doi.org/10.2307/1912791>; see also Risser et al., 2025 <https://doi.org/10.1088/2752-5295/add046>), although the nonstationarity of both the covariates and response variables are problematic. Whether you are inferring causation or correlation will be very important if the results are ever to be used in court for loss and damages! Therefore, I think you need to be *very* clear about the nature of the attribution statements you're making, and what sort of causality is implied.

(Remarks on code availability)

Version 2:

Reviewer comments:

Referee #5

(Remarks to the Author)

Dear authors,

Thank you for your careful evaluation of and response to my comments. I am now happy to recommend the paper be accepted for publication.

(Remarks on code availability)

Response to Referees 1-4 for the manuscript

Systematic attribution of heatwaves to the emissions of carbon majors

Yann Quilcaille¹, Lukas Gudmundsson¹, Dominik L. Schumacher¹, Thomas Gasser², Richard Heede³, Corina Heri⁴, Quentin Lejeune⁵, Shruti Nath⁶, Philippe Naveau⁷, Wim Thiery⁸, Carl-Friedrich Schleussner^{2,9}, Sonia I. Seneviratne¹

¹Institute for Atmospheric and Climate Science, Department of Environmental Systems Science, ETH Zurich, Zurich, Switzerland.

²International Institute for Applied Systems Analysis (IIASA), Laxenburg, Austria.

³Climate Accountability Institute, Snowmass, Colorado, USA

⁴University of Zurich, Zurich, Switzerland

⁵Climate Analytics, Berlin, Germany

⁶Atmospheric, Oceanic and Planetary Physics, Department of Physics, University of Oxford

⁷Laboratoire des Sciences du Climat et de l'Environnement, ESTIMR, CNRS-CEA-UVSQ, Gif-sur-Yvette, France

⁸Vrije Universiteit Brussel, Department of Water and Climate, Brussels, Belgium.

⁹Integrative Research Institute on Transformations of Human-Environment Systems (IRI THESys) and the Geography Department, Humboldt-Universität zu Berlin, Berlin, Germany.

Correspondence to: Yann Quilcaille (yann.quilcaille@env.ethz.ch)

We would like to sincerely thank the four Referees for their constructive feedbacks. We have carefully accounted for all comments, leading to a significant improvement in the quality of this manuscript.

We detail how we address the comments of each Referee consecutively, with the modifications brought to the manuscript. Referees' comments are shown in **black**. The authors' response is shown in green text. The text quoted from the manuscript is shown in **blue** between quotation marks in italics. Here is the summary of the modifications:

- Better explanations to detail that this manuscript builds on existing publications, and how/why it differs. Providing this context underscores that this manuscript uses very well-established methods.
- Reworked the section on the decomposition of the probabilities into contributions for the carbon majors. We acknowledge that the explanations were lacking in clarity and rigor because we attempted a technical discussion for the broad audience of *Nature*. We have included an adequate expert, Philippe Naveau, to work on this part. We reframed this section, now in Supplementary Information, showing numerically that our approximation is valid for all but one heatwaves.
- Various clarifications in the text and improved figures, with new ones in Supplementary.
- Improvements on the Discussion: less repetitions from the manuscript, but more on missing emissions & implications of the results for climate litigation.

For every comment, the modified text is written if feasible. However, some comments required a large number of modifications, and listing all of them would considerably lengthen this

response. For the sake of conciseness, we list the number of the lines modified accordingly to the version with tracked changes.

Contents

Response to Referee #1 3
Response to Referee #2 8
Response to Referee #3 11
Response to Referee #4 13
References 22

Response to Referee #1

Referee #1 Comment 1

The article conducts a systematic event attribution of 187 historical heatwaves, and attributes the effects to individual carbon majors. The novelty lies in the fact that this is such a large systematic study and the results are therefore comprehensive and relevant to a wide range of readers, the attribution to carbon majors is also an interesting development. As such I think that this paper is worthy of being published in this journal.

Response to Referee #1 Comment 1

We would like to sincerely thank Referee #1 for this positive summary.

Referee #1 Comment 2

My main concern is that at times it is hard to determine what is new here, I think this would be the case for the many non-specialist readers of this journal . For example which of the methods being employed are themselves novel and which are identical or very similar to those already being used in other studies. While many of these studies are referenced and an expert or interested reader could discover this themselves I think it would be helpful to include more phrases such as “following ...” or “similar to ..” to make it very obvious when an existing method is being used.

Response to Referee #1 Comment 2

Thank you very much for identifying this important and constructive feedback. Many readers are indeed not familiar with the usual approaches in the field, and it should be taken into account. We have brought many modifications throughout the Main Text and the Supplementary Information to point out what is new and what is usual in this work with respect to the rest of the literature. Namely, we have made explicit when the method is strictly the one of the WWA, or when and how it differs. Additionally, we clearly how this analysis is similar or different from the existing studies on the assessments of contributions to global climate change, the works of Otto et al, 2017 and Lott et al, 2021, and the four studies assessing the contributions of the carbon majors.

The modifications occur over the lines: L23-24; 64-65; 69-70; 84-85; 88-91; 93-95; 102; 166-175; 268-271; 301-302

Referee #1 Comment 3

Reading the section beginning L44 might also give the impression that this is the first study to attempt to attribute extreme events to individual sources – so care needs to be taken to ensure that the reader realises that there are already several studies linking extreme events to countries (e.g. Otto et al 2017) and individuals (e.g. Lott et al 2021).

Response to Referee #1 Comment 3

It was indeed insufficiently explained in the introduction that these two studies already linked climate extremes to emitters. Following the former comment, we are now clearly writing that Otto et al, 2017 and Lott et al 2021 already advanced this link in the introduction (L50-51; 54-55; 57-58; 69-70) and in the main text (L168-169; 172-174).

Referee #1 Comment 4

The article should also be clearer if this attribution is to anthropogenic effects or climate change in general as it seems to be described both ways e.g. L24 vs L113.

Response to Referee #1 Comment 4

Thank you for pointing this lack of clarity. The first part of this work attributes the heatwaves by comparing the distributions at the GMST at the time of the event and at preindustrial, thus includes all effects. As such, we have clarified the whole first part to write that we are attributing to climate change in general, not anthropogenic effects (L21; 27; 47; 50; 66; 97; 98; 100; 125; 135; 145; 156; 160). In the second part, the contributions are attributed to the emissions of the carbon majors. As such, we have clarified that we evaluate the contributions of anthropogenic actors, and “*other unidentified contributors, anthropogenic and natural*” (L227-230).

Referee #1 Comment 5

In several places you describe how climate change has increased the risk and amplitude by a certain amount but do not specify from when (I assume this is 1850-1900) but it would be good to be more explicit, especially in the abstract.

Response to Referee #1 Comment 5

Thank you for pointing this out. We acknowledge that this information was not sufficiently detailed. We have made clear that the reference period is indeed 1850-1900 in the abstract, in the text and in captions (L28; 34; 98; 104; 136; 152; 162; 222; 240; 288).

Referee #1 Comment 6

The values for attributable risk seem quite high. For the example presented for the Pacific NW heatwave, the previously calculated value of 150 times on L96, although consistent with your uncertainty, is much smaller than your median (over 10000). Can you comment on this and perhaps give an example of such large numbers being calculated in previous literature?

Response to Referee #1 Comment 6

Thank you very much for this comment. In this response, we will first discuss the Pacific NW heatwave of 2021, then discuss large PR in the literature.

Regarding the Pacific NW heatwave of 2021, the former work¹ calculated a PR of *at least* 150. We have changed in the text from “*higher than*” to “*at least*” (L111) for added clarity. When looking into the details, Figure 12 of Philip et al, 2022 shows a probability ratio with the lower bound of the confidence interval of 150 and the upper bound higher than 10,000. Our analysis shows that the lower bound of the confidence interval, 7, is below the one of Philip et al, 2022. Both studies show a higher bound higher than 10,000. Finally, our median is higher than 10,000. Philip et al, 2022 does not report the synthesis on the median, but Table 2 gives insights. While ERA5 has a median of ~350, 11 ESMs out of 21 have a median higher than 10,000. Another work analyzes the Pacific NW heatwave². Although it does not communicate its results on the probability ratio itself, it does inform about the return periods, which are consistent with ours.

Thus, these differences that we obtain for the Pacific NW 2021 event can occur due to an ensemble of reasons:

- Characterization of the event: we use here the reported US component of the PNW 2021 event, yet this event was mostly over Canada, and Philip et al, 2022 uses a box over both countries. Besides, we use the average over the period, while Philip et al, 2022 uses the maximum temperature.
- Another reason for this higher value is that unlikely events have larger uncertainty ranges, and more subjective to definition choices, as shown in another work on the PNW 2021³. However, Zeder et al, 2023 does not precise the values of their probability ratios, preventing a direct comparison. Nevertheless, we have discussed in more details this limit in the text.

We have modified the text to explain these reasons (L112-113; 117-120). The modified text is as follows:

“Climate change has also increased the probability of the heatwave at least 7 times, although the median estimate and the upper bound of the confidence interval are higher than 10,000. This attribution is consistent with existing works on this heatwave: a first work¹ found a probability ratio of at least 150 and a change in intensity of 2.0 °C (1.2 to 2.8), while another analysis³ suggested more than 2.9 °C. Although consistent, the results differ due to the characterization of the event and its very unlikely nature. The region and period are here determined through the reporting of the disaster rather than arbitrary choices. Choosing the maximum temperature as indicator over the period amplifies the extremeness of the event^{3,4}. Very unlikely events like the Pacific Northwest heatwave of 2021 are more difficult to investigate, increasing the dispersions across several analysis⁵. However, this increased dispersion does not lower the confidence in the conclusion, which is the strong influence of climate change on such unlikely events. The method described for this event is applied to the 187 heatwaves, with three other cases showcased in Figure 1c-e, with the results also consistent with available attribution studies.”

Referee #4 asks whether very high values for the PR may be found in the previous literature. There are other studies that find heatwaves with probability ratios that high and even higher, such as the prolonged heatwave in Siberia in 2020⁶. This study finds that the ratio is of at least 500, and a median estimate of 90,000. Sadly, this heatwave is not among the two Russian heatwaves provided by the EM-DAT database, preventing a comparison.

For information, it might be possible that the method to aggregate multiple lines of evidence, i.e. synthesizing the observations and models, could also affect the results. We use the simple approach proposed by the WWA (see the last paragraph of Methods/Evaluation of the uncertainties). Other methods could be used, but investigating this effect should represent a dedicated journal. Nevertheless, we mention this limit in the aforementioned paragraph in the Methods for the sake of clarity (L413-414).

Finally, the decadal numbers of Figure 2 have marginally changed in this round of review. There are two reasons. First, we are now using the latest version of ERA5, allowing us to fit the statistical models up to 2023 instead of 2022. Then, when doing a thorough check, we noticed that for some events, the majority of climate models were rejected according to the criteria on the scales & shapes parameters (see next comment). It implies that we had less than the target of 10 used ESMs. Having too few ESMs may reduce the stability of the estimate for climate models. We have decided to adjust the criteria of selection as follow. If there are enough ESMs that respect the criteria on the scales & shapes parameters, the ESMs will be ranked based on the local seasonality of ERA5, and the 10 best ones are selected. If there are not enough, this list will be partially complemented with the best remaining ESMs according to the local seasonality of ERA5. This second case has only marginally changed the results.

Referee #1 Comment 7

L308 – could you give a bit more detail on why this is the best method for model selection – or give a reference where this is also done – as its not immediately clear to me why a model

with good seasonality would be best at modelling the frequency of extreme events.

Response to Referee #1 Comment 7

Thank you very much for this insightful comment. In a nutshell, our method relies on two steps:

1. The first step recommended by the WWA is to check the capacity of numerical models to reproduce observed climate, here the local seasonality of ERA5.
2. The second step corresponds to a statistical goodness of fit, which insures that the statistical distribution is adapted to the dataset at hand. Besides, the scales and shapes parameters of the fits trained on climate models are compared to the parameters of the fit trained on ERA5, ensuring similar distributions.

Yet, as acknowledged by the WWA^{7,8}, the first round is necessary. To use the former example, the PNW 2021 analyzed by Philip et al, 2022¹ compares the seasonal cycle and the spatial pattern in their Table 1, based on the method described in the supplementary material of Ciavarella et al, 2021⁶. There, the seasonal cycle of the monthly maximum over the region of interest is also compared between observations and models. In our case, we compare the seasonality of the daily average temperature, because we use the average over the period of interest and not the maximum. Similarly, we use the seasonal cycle over the full year to assess whether these models represent well the climate of the region. Once the performance is calculated in every grid point, it is averaged over the region of interest, as an approach to assess the spatial pattern, instead of comparing the seasonality over the average of the region.

We acknowledge that our method differs slightly from the one described in Ciavarella et al, 2021⁶. These differences are meant to facilitate the systematic analysis of the whole set of heatwaves, and we consider them minor. These technical details are already entirely described in the section Methods (L398-407), thus will be accessible to all readers.

Referee #1 Comment 8

Minor comments:

Figure 1 – could you make the outline of the events a bit clearer.

Response to Referee #1 Comment 8

It was indeed hard to read. Figure 1 is now modified, we hope that it looks better.

Referee #1 Comment 9

L277 – perhaps change “former” to “previous” as I think this would be clearer.

Response to Referee #1 Comment 9

Good point, changed.

Referee #1 Comment 10

L284 – please explain what NLL means – i.e. negative log likelihood.

Response to Referee #1 Comment 10

Sorry, we used an acronym while it is not needed. Replaced with negative log likelihood.

Referee #1 Comment 11

Could you consider changing GMT to GMST – as I think this is more commonly used – this is also the abbreviation that is used for this metric by the IPCC

Response to Referee #1 Comment 11

Thank you for this suggestion. It has been changed in the Main Text, Methods, Figure 1 & 3, and Supplementary Information (L95; 131; 143; 195; 205; 217; 218; 224; 225; 236; 367; 372; 396; 397).

Referee #1 Comment 12

Referee #1 (Remarks on code availability):

Unfortunately I have run out of time to try to get this code to actually work (its quite complicated so this would take longer than I have).

However I have looked through it and can confirm that there is a README file which looks much more helpful than most I've seen and the code itself is well structured and commented so it all looks good to me.

Response to Referee #1 Comment 12

We want to sincerely thank you for making such an effort. We are aware of much time can be invested in trying to run another code in a review. In this case, there are a lot of data to download and prepare (CMIP6, ERA5, BEST, GADM, EM-DAT). Even if we did provide the code to prepare the data, run, analyze, and plot everything, it remains a lot. We have sorted and documented everything, but we agree that it can always be improved, especially for other users.

For information, the code for this manuscript is not only stored on Zenodo, but it is also on a GitHub repository that we will maintain.

Response to Referee #2

Referee #2 Comment 1

General comment:

The authors present a systematic approach to attribute altered heat wave risks to the contribution of individual polluters, commonly referred to as carbon majors.

[...]

Sure, most events and most companies have much smaller footprints, but the fact that these contributions can be assigned to individual polluters is - in my view - a milestone. The attributed change in risk of occurrence is equally interesting, though less intuitive for the target audience.

Summary:

As you can certainly tell from my rather enthusiastic general review comment (quibbles thereafter), I wholeheartedly recommend the paper for publication in Nature given its thoughtful state-of-the-art approach and analysis, as well as its careful consideration of the available observational data and modelling tools. In other words, quality of the data as well as the representation is excellent. Context and clarity is provided (with a few exceptions raised below). The authors convincingly demonstrate that the results are robust within the bounds of widely accepted uncertainty analysis standards. As a note of caution, I will say that Bayesian statistics is not my strong suit (to put it mildly), hence I do put a fair amount of trust in them applying the respective methods (Bayes' theorem and the Inclusion-Exclusion principle) with best intentions and due diligence. Their discussion of the caveats and the overall impression of the presented manuscript makes me very confident that the analysis stands the test of better stats experts than I am. That all said, I do have a few questions that I would like the authors to be addressed before possible publication.

Response to Referee #2 Comment 1

We deeply appreciate your support to our study and very much share your enthusiasm. Taking the time to recognize the hard work as you did here is always very encouraging for the authors. Thus, I and all the co-authors want to sincerely thank you.

Referee #2 Comment 2

Minor comments and quibbles:

Lines 202-204 (and 231-233): "The heatwaves over 2000-2022 had their intensity increased on average by 1.59 °C due to climate change, of which 0.40 °C is due to the seven top carbon majors, and 0.41 °C is due to the 115 others."

—> It took me several times reading this, until I figured where it's probably coming from. Am I correct to assume that it is the average of the 2000-2009, 2010-2019, 2020-2022 intensity increase (of all 187 heatwaves) presented in the section above? What leaves me still puzzled is the fact that you refer to the decadal results (1.3°C, 1.6°C, 2.0°C, resp) as median average, where here it's just average ... so what is it? Also, I suppose that the 0.81°C for all 122 carbon majors is the result based on the OSCAR model? Please clarify and try to improve the manuscript at this point in order to make it easier for other to follow.

Response to Referee #2 Comment 2

Thank you for pointing this out. The decadal results are indeed based on medians, while we used the average of the change in intensities over the whole period here. We have decided to use the decadal medians here as well. Doing so has shown that the contributions of carbon

majors in heatwaves is increasing over time, which we had missed initially. The text now reads as follows:

“For comparison to the influence of climate change, we calculate the influence of carbon majors over heatwaves reported over each decade of our dataset. With reference to 1850-1900, climate change has increased the median intensity of heatwaves by 1.34 °C over 2000-2009, of which 0.35 °C is traced back to the seven top carbon majors, and 0.30 °C to the 115 others. These contributions correspond respectively to 26% and 22% of the overall effect of climate change. Over 2010-2019, the influence of climate change increases to 1.61 °C, with 0.40 °C (25%) from the seven top carbon majors and 0.43 °C (27%) from the 115 others. Over 2020-2022, it increases to 2.08 °C, with 0.53 °C (26%) for the seven top carbon majors and 0.59 °C (28%) for the 115 others. These results show that carbon majors’ emissions contribute for about half of the increase in intensity of heatwaves since pre-industrial times, and that this contribution is rising.”

Referee #2 Comment 3

The discussion section could be made a lot more exciting. To be honest, it felt a bit flat after reading through an otherwise great paper. It contains a lot of (partially literal) repetition of what was just said in the previous paragraphs. Feels a bit of a waste given the strict space limitations. For example, the legal repercussions of the findings could be discussed, but I could also do with an expansion from a caveats perspective. One concrete aspect in terms of caveats is the impact of the counterbalancing (i.e. cooling) aerosol effect, that may enhance the carbon majors warming contribution even further. It’s briefly mentioned in lines 242-247, but perhaps there’s room to assess things especially in terms of qualitative aerosol impact estimates.

Response to Referee #2 Comment 3

Thank you for this constructive feedback. We have rewritten the first two paragraphs of the Discussion to avoid repetitions (L268-300). We have also expanded further the discussion following this recommendation.

We were aware of the legal implications of this work, but we were concerned that such scientific evidence may be dismissed in a court of law if apparently motivated by legal concerns. This is why we had reduced this opening to the bare minimum. Nevertheless, because our motivation remains purely scientific, we can follow this recommendation while acknowledging Referee #3’s perspective. We have acknowledged explicitly the potential of this work as scientific evidence in the last paragraph of the Discussion, while remaining careful (L333-349).

Regarding aerosols, we have also developed this part. We have detailed further the effects of missing emissions, both in aerosols and CO₂ emissions (L301-324). An existing work⁹ has allowed to estimate the combined effects of missing emissions, although it remains an approximation.

Referee #2 Comment 4

Line 211: What are CNX resources? Acronym not referenced.

Response to Referee #2 Comment 4

CNX Resources is one of the carbon majors. As far as we could find, this company has not officially defined its acronym. We have therefore detailed the sentence to mention that it is a carbon major, to avoid confusion (L259-260).

Referee #2 Comment 5

Figure 1: Would it make sense to have the same return period interval on the bars on the rhs in each of the 4 station panels? It's obvious why it's different for temperature, but not why it's different for the return period labelling. Also, is that same figure available for BEST (to be put in the supplementary material)?

Response to Referee #2 Comment 5

Thank you for the suggestions. We have modified the ticks for the panel on return periods, so that they match across events. Readers will now have a better capacity to interpret the likelihood of the events, while seeing how far it is from the distributions. Regarding the figure with BEST, it is a good idea as well, we have done so.

Referee #2 Comment 6

Referee #2 (Remarks on code availability):

Two of my colleagues and I did take a look at the code and found it very well explained and structured. Given that a massive amount of CMIP6 data need downloading before the scripts can be run and tested, we did not have the time to do that. So I do not know whether this counts as having reviewed the code, but I reckon it might, so I put 'yes'.

Response to Referee #2 Comment 6

Thank you very much for this feedback. For information, we will update the scripts on the Zenodo & GitHub with all modifications brought in this review.

Response to Referee #3

Referee #3 Comment 1

I have been asked, if the findings of this paper, are technically robust, whether they would have important implications for legal and/or regulatory actions.

This paper makes an important contribution to the growing literature on climate attribution. There are now approximately 25 lawsuits pending in the courts in the United States, and a small number in other countries, where climate attribution would be relevant. The suits typically seek money damages from the fossil fuel majors for the costs of adaptation to climate change. A few of these cases have been dismissed. The others were held up for years by various jurisdictional questions. None of them has gotten to trial or even to discovery on the merits, and thus to issue of attribution has not yet been litigated.

One of the pending cases was brought by the City and County of Honolulu against several fossil fuel majors. The Hawaii Supreme Court ruled for the plaintiffs. The defendants have asked the U.S. Supreme Court to take the case. The Supreme Court has asked the Solicitor General of the U.S. for her views on whether they should take it; she has not yet responded. This inquiry shows that the Supreme Court may be interested in taking the case. If they do take the case, it is entirely possible that they will issue a ruling that has the effect of wiping out all 25 of these cases. If that happens, this paper may be of little legal importance in the U.S. However, if the Supreme Court refuses to take the case, or if it takes the case and affirms the holding of the Hawaii Supreme Court, then this paper would be of great significance. I expect that it would be cited by the plaintiffs in most or all of these cases. The defendants will then try very hard to discredit it. Thus thorough technical peer review is especially important.

The paper is also relevant to international negotiations on the issue of loss and damage — how much money should the developed countries pay the developing countries to compensate them for their losses from climate change. There may be no judicial remedy to seek loss and damage, and ultimately the developed countries (or some of them) may refuse to pay significant amounts, but this paper will certainly be of interest to the negotiators from the climate-vulnerable countries.

Response to Referee #3 Comment 1

We would like to thank you for your insightful perspective on this manuscript. We are aware of the legal implications of our work, and are committed to ensure its scientific value.

The first point that we strived on was our scientific objectivity. We were worried that giving too much voice on the application to climate litigation may be perceived in a court of law as a biased article. Yet, our objective was and remains to provide robust scientific information, filling in gaps in climate science, although it does provide resources as scientific evidence. Given this perspective and a recommendation from Referee #2, we now explicitly acknowledge the potential of this work as scientific evidence in the section Discussion as follows.

“These results are relevant not only in the scientific community but also for climate policy, litigation and wider efforts concerning corporate accountability^{10,11}. Climate-related legal proceedings are proliferating, with defendants seeking compensation for losses and damages or requiring more ambitious climate actions from corporations and nations^{11,12}. However, the

scientific evidence backing the claims are often lagging behind the state of the art in climate science, thus failing to adequately draw causality links¹³. Although this work aims at filling in scientific gaps, the results also fill in evidentiary gaps. This systematic attribution improves the coverage in extreme events, thus reinforcing the potential of attribution science for climate litigation^{13,14}. Furthermore, if the fact that fossil fuels are the main driver of climate change has been unambiguously established^{15,16}, as acknowledged by the carbon majors themselves¹⁷⁻¹⁹, quantifying the causality from the emitters to the events provides important new resources to assess legal responsibilities. Strengthening further the links between climate scientists, legal scholars and practitioners will be beneficial to ensure that the overwhelming scientific literature is correctly accounted for²⁰.”

The second point corresponds to the scientific robustness of this work. We fully agree that the high potential of this paper would likely cause the defendants to attempt to dismiss the value of this work. We had already strived on the quality of this work, as acknowledged by Referee #2. We think that this process of peer review has improved further the robustness of this work, thus reducing its risk of being discredited. Four Referees have provided feedbacks on this manuscript, and all have been useful, from the very critical perspective of Referee #4 to the very enthusiastic perspective of Referee #2.

All comments have given us excellent opportunities to improve the work on key aspects, and limit the risks of defendants discrediting this paper. Referee #4 has been very critical even on well-established methods heavily backed by scientific literature. However, Referee #1 pointed out that the manuscript was not sufficiently clear on which existing methods or papers this work builds on. Now, we clearly acknowledge former works, referencing which well-established methods this paper builds on, how it differs and justifying why. We are convinced that it will help the readers understand how this work fits in a broader context, but also of its high scientific value. Referee #4 was also critical on the probability calculus: this part was novel, and not correctly explained. We have reconsidered our approach, thanks to an expert (Philippe Naveau) invited as co-author. We consider that the new version of this manuscript is robust enough to stand the efforts at dismissing the conclusions of this paper.

As Referee #3 points out, the legal importance of this work depends not only on our work. If the U.S. Supreme Court takes the case, and rules them out, then this work would be less relevant for climate litigation in the U.S. Yet, lawsuits are flourishing not only in U.S. courts of law, but also in other national or even international courts^{11,21}. Nonetheless, while we would support the use of this work as scientific evidence for climate litigation, its first and foremost purpose is for science.

Response to Referee #4

Referee #4 General comment

The manuscript presents an analysis of 187 historical heatwaves from 2000–2022 and attempts to attribute these events to carbon emitters and anthropogenic climate change. While the topic is highly relevant and timely, the statistical methodologies employed raise several critical issues, particularly concerning accuracy, rigor, and adherence to fundamental principles of probability theory and extreme event attribution. In its current form, the conclusions drawn from the analysis appear unreliable due to incorrect model assumptions and flawed statistical methods.

For the reasons detailed below, I recommend rejecting the manuscript. A substantial revision of the methodology and analyses is necessary before considering publication in any outlet, especially a journal as prominent as *Nature*.

Response to Referee #4 General comment

We would like to thank Referee #4 for offering their perspective. We believe that the very critical comments have been instrumental in improving the manuscript. Referee #4 criticized two aspects of the analysis, namely the link to the carbon majors and the extreme event attribution (EEA) itself.

- In response to Referee #4's comments, we have realized that there were imprecisions in the presentation of our approach that have contributed to misunderstandings of the actual approach in our work. We would like to apologize that these unclarities on our side have led to confusion and have revised our approach to increase clarity and transparency. In the process of revising our approach, we have invited a world leading expert on extreme events, statistics of climate change, and probabilistic modelling, Philippe Naveau, to the author team, to assist with improving our approach. We are confident that the outcome has the necessary scientific quality to be published in a journal such as *Nature*, while remaining accessible to a large audience.
- Fundamentally, we note that our work is based on very well-established methods, in particular the probabilistic approach employed by the World Weather Attribution (WWA). Listing all the articles using this approach would be hard²², but as a non-exhaustive list, we can mention the following articles in the *Nature* portfolio^{2,6,23-32}, in the *Science* portfolio^{33,34}, in various journals of high quality^{1,35-54}, as well as many other journals, even in other fields such as *the Lancet*⁵⁵. Referee #1 has rightfully pointed out that this manuscript did not sufficiently detail which approaches it builds on, how it differs, and justify why. We have now clearly referenced and explained these aspects in the manuscript. We would also like to point out that Referee #2 praises our analysis for its quality and the use of a state-of-the-art approach.

Referee #4 Comment 1

1. Attribution of heatwaves to carbon majors

Response to Referee #4 Comment 1

In the former version of the manuscript, the objective of this part was to justify the existence of the decomposition into a sum of terms, explain that the exact calculation of each term isn't feasible, and thus propose an approximation. However, the explanation was not rigorous and clear enough, which led to several misunderstandings.

In the new version of this manuscript, we provide a more rigorous definition of the problem, explain that this problem can be approximated in a Gaussian case, but that the use of

Generalized Extreme Value (GEV) distribution hinders our case. Thus, we propose an approximation. While an analytical demonstration of the approximation remains very challenging, we verified numerically the validity of this approximation for each heatwave. This section has been entirely moved from the Methods to the Supplementary Information, so that we can use figures to showcase our approach.

Referee #4 Comment 1.1, part a

(1.1) The treatment of events and probability is very not rigorous. What is the probability space S under consideration? What is its associated σ -algebra B ?

Response to Referee #4 Comment 1.1, part a

Due to the broad audience, we had decided to avoid introducing concepts unfamiliar to the vast majority of readers, such as a σ -algebra. However, following this recommendation, we have explicated that it is here a Borel σ -algebra on the reals, as detailed in the Supplementary Information.

Referee #4 Comment 1.1, part b

(1.1) The definitions of R and N as factual and counterfactual conditions, respectively, are imprecise. In probability theory, if N represents “world that might have been” conditions that never happened, $P(N)$ should be 0, while $P(R) = 1$, as the factual conditions did occur. The assumption of $P(N)/P(R) = 1$ (see Line 176 of the Supplementary Material) (Line 176 of the Supplementary Material) is inconsistent with basic probability principles.

Response to Referee #4 Comment 1.1, part b

We acknowledge that we did not explain well the technical setup. The Earth system can produce different climatological distribution under different forcings, say A and B . Even if $P(A \text{ and } B) = 0$, this does neither imply that $P(\text{Event}|A) = 0$ nor $P(\text{Event}|B) = 1$ (or vice-versa). Similarly, although an individual cannot be a smoker and a non-smoker at the same time, the probability of having lung disease given the individual does not smoke is not zero. So, we did not explain that factual and counterfactual are conditional features of our system. To avoid confusion, we remove this discussion in the new version, and we refer to Hannart et al. (2016)⁵⁶ for a detailed explanation of counterfactual theory in climate sciences.

Referee #4 Comment 1.2

(1.2) In probability theory, if we have two events N and R in a probability space (S, B) such that $N \subset R$, then any occurrence $s \in N$ will satisfy $s \in R$, which means R must happen when N happens.

However, by definition, the counterfactual/unperturbed conditions N cannot happen simultaneously with the factual conditions R , which is the basis principle of extreme event attribution or causal inference. That is to say, $R \cap N = \emptyset$, which directly contradicts Equation (2) on Line 373.

Response to Referee #4 Comment 1.2

We apologize for causing this severe misunderstanding. The conditions described with N and R were meant to describe drivers of climate change. On one hand, R corresponds to a factual scenario, with both natural and all anthropogenic drivers affecting the Earth system thus the probability of the heatwave. On the other hand, N corresponds to a counterfactual scenario, with only natural drivers, thus with a different probability of the heatwave. Thus, while R is occurring, all drivers occur, including the natural ones that are entailed by N .

This part is now removed to avoid any confusion in this section.

Referee #4 Comment 1.3

(1.3) A union of two events $E_1 \cup E_2$ indicates either E_1 or E_2 happens, not requiring both events to happen at the same time. However, for the factual conditions R to occur, all anthropogenic activities, including O and $E_i, i = 1, \dots, n$. Therefore, should co-occur. Thus, the correct formulation of R should involve intersections:

Response to Referee #4 Comment 1.3

N, R, O and E_i are not events, but are drivers. Sorry for having created this confusion. This part is now removed to prevent misunderstandings.

Referee #4 Comment 1.4

(1.4) To derive $\delta PR_{X,E_i}$, the Equation (12) in the Supplementary Material requires simulating and computing [...]. Considering there are carbon $n = 122$ majors under consideration in this paper, running the OSCARv3.3 driven by $2^{122} - 1$ different counterfactual scenarios is computationally infeasible. The authors should clarify how they managed this massive computational burden.

Response to Referee #4 Comment 1.4

Referee #4 is right in pointing out that it is not feasible to calculate that many counterfactual scenarios. This is why we had introduced an approximation, but that was insufficiently explained. The well-established method to calculate contributions to global warming is to use perturbations⁵⁷⁻⁶⁷. This method proposes to calculate not all combinations, but only the factual scenario and as many counterfactual scenarios as there are contributors, and to deduce their contributions as the difference between factual and counterfactuals. We agree that this method remains an approximation, although it has been shown to provide highly satisfying results⁵⁷⁻⁶⁷.

Acknowledging the high non-linearities in the probabilities, we had proposed an improved approximation, described L394-404 in the original manuscript. The principle is to average the traditional term obtained by removing one contributor to a new term obtained by adding just a single contributor to a counterfactual without them. In the new version of the manuscript, this approximation is explained in more details in the Supplementary Information. We show that this new improved approximation leads to better results. We acknowledge that for a single case of all 187 events, the approximation is shown not to be good enough, which is why we have removed this event from the analysis. This case is explicitly mentioned in the supplementary information.

Referee #4 Comment 1.5

(1.5) Clear communication of causal assumptions and the interpretation of conclusions is essential in climate change detection and attribution studies. Attribution results based on Earth System Models (ESMs) are inherently subject to structural uncertainties, and their reliability depends heavily on the model's ability to accurately simulate the relevant variables of interest (Wehner, 2023).

Response to Referee #4 Comment 1.5

We agree that Earth System Models (ESMs) have inherent uncertainties, and more importantly different regional performances. This is a prominent issue for extreme storms as shown in Wehner, 2023⁶⁸, and although still applicable⁶⁹, is less problematic for heatwaves⁷⁰⁻⁷². This is precisely the reason why the protocol of WWA includes both observations and models, but restricts analysis to validated models, using the best available observations (or

observation-derived estimates) as a reference. Specifically and following this protocol, we select the ESMs based on their ability in reproducing the regional seasonality as well as the parameters of the trained statistical distributions, using ERA5 as a benchmark. Together, this ensures that the seasonal dynamics of temperature as well as the variability and tail behavior of an extreme event index, in our case mean temperatures during heatwaves, are adequately captured for a given analysis region. Overall, we are adequately selecting the ESMs appropriate for each event. Referee #1 had an insightful comment on L308, and we encourage Referee #4 to check our answer. Additionally, more details are provided in the protocol of WWA^{6,7}.

For instance, in the case of the June 26–29, 2021 Pacific Northwest (PNW) heatwave, Bercos-Hickey et al. (2022) employed ensembles of **regional** climate models, including the Weather Research and Forecasting (WRF) model and the International Centre for Theoretical Physics Regional Climate Model (RegCM 4.9.5). These simulations, initialized on June 22, 2021, used historical boundary conditions from the Global Forecast System (GFS) analysis and ran continuously through July 2, 2021. Despite the use of regional models specifically designed to capture *localized* climate dynamics, the highest temperatures simulated by WRF were still approximately ~ 1 °C lower than observed in the PNW.

Given this, it raises doubts as to whether a reduced-complexity **global** ESM like OSCARv3.3 can adequately reproduce the intensity of all 187 heatwave events analyzed in the study. Indeed, previous research has shown that global climate models frequently underestimate the intensity of extreme heat events (see e.g., Stott et al., 2016), further questioning the robustness of the model-based conclusions in this paper.

We apologize for any confusion that we may have caused, but it seems that you misunderstood the method presented in this paper. OSCARv3.3 does not calculate the intensity of the heatwaves but rather provides the Global Mean Surface Temperature (GMST). OSCAR is a climate emulator like MAGICC, HECTOR or FaIR, and thus does not have the capacity to emulate heatwaves. Instead, our method is in two steps:

- Apply the well-established approach of the WWA that combines observations and selected ESMs for the attribution analysis, providing us with statistical models driven by GMST.
- Then, OSCAR calculates contributions to GMST from the carbon majors, that we combined to the statistical models. In the original manuscript, it was described on L161-169 & 323-363. However, we have expanded this part to provide a clearer and more comprehensive understanding.

Regarding the uncertainties & model biases in OSCAR, the model is run with observational constraints (see Methods). Furthermore, the statistical model is always established/fitted based on the original GMST from ERA5 and ESMs, respectively, to ensure consistency. Using reduced-complexity ESMs to assess contributions to global warming is a well-established approach⁵⁷⁻⁶⁷, and as discussed in the next section, using the approach of WWA to link GMST and heatwaves is also a well-established approach^{7,22,73,74}. We hope that these clarifications make it clear how we establish the full chain from carbon major to individual extreme events.

Referee #4 Comment 2

2. Attribution of heatwaves to climate change

To study the effect of anthropogenic climate change on the intensity of a certain heatwave event, the authors took a spatial average of the daily temperature within the impact region of

the heatwave to obtain an annual time series. Then they assumed the spatially-averaged temperatures subject to a Generalized Extreme Value Distribution (GEV) distribution.

They claimed that this type of analysis applied individually to 187 different heatwave events is systematic, which I can hardly agree with. Also, their practice violates the classic Extreme Value Theory (EVT), and also the inclusion of Global mean temperature (GMT) as a covariate in the location parameter of the GEV distribution is somewhat naïve.

Response to Referee #4 Comment 2

As previously mentioned, we are surprised by the unequivocal dismissal of well-established attribution methods, which have proven their potential in a very large number of publications²², including articles in various high-level journals such as *Nature* and other^{1,2,6,23-54}.

Further improving this method following your suggestions would always come with drawbacks as we explain further below^{69,75}, and we also explain how such modifications would come at the cost of inadequately addressing our research question. We discuss why a GEV remains a correct approximation, why we choose the GMST covariate, and rebut the other points in the corresponding comments.

Referee #4 Comment 2.1 part a

(2.1) For the GEV assumptions (1) to hold, one has to work with block maxima with a block size $n \rightarrow \infty$, which is why Statisticians usually fit GEV distributions to annual maxima or seasonal maxima. Also, it is important to make sure that the observations within a block $\{Y_1, \dots, Y_n\}$ all subject to the same distribution. Considering the daily temperature in a year has autocorrelation and seasonality among days, the records are not identically distributed. The authors must conduct a GEV goodness-of-fit test to their spatially-averaged records because the GEV assumptions have all been seriously violated.

Response to Referee #4 Comment 2.1, part a

We acknowledge that using a GEV for heatwaves is an “assumption”, or rather an approximation. As Referee #4 mentions, annual/seasonal maxima are fitted by a GEV, but this remains an approximation because of the finite block size. As approximations, GEV or any type of distributions should be only viewed as possible candidates to fit the data at hand. Because statisticians are widely using a GEV as acknowledged by Referee #4, it should not be an issue for us either.

The essential statistical question is to determine if the distribution of interest, here the GEV, may provide (or not) an adequate fit to your data, especially for large values. To settle this point, we compared different distribution types, and the GEV outperforms the others. The demonstration is provided at the end of this response. If the editor thinks that it is needed, it could be added to the Supplementary Information of this manuscript.

Referee #4 then refers to the temporal correlations. We remark that the points of our sample are annual values, e.g. the average over June 26-30 in every year for the Pacific Northwest Heatwave of 2021. There are low interannual correlations between such events, except in some situations in the tropics due to ENSO. Furthermore, we acknowledge that we are not using a block maxima over the period, but the average. There is a plethora of metrics for heatwaves⁷⁶, and we have chosen this one as a middle-ground. Additionally, epidemiological studies often rely on mean temperatures⁷⁷⁻⁷⁹. Event-averaged temperatures have already been employed by a very large number of attribution studies, such as^{7,22,73,74}.

While using a GEV was already heavily backed by the scientific literature on EEA, we had already discussed this part in the Supplementary Information L53-89. To summarize, we tested for each heatwave which distribution led to the best results in terms of Bayesian

Information Criteria. We conserve this section, which shows that the GEV is indeed the best approximation.

Not only is the use of the GEV distribution already heavily backed by the scientific literature on EEA, and justified in following paragraphs, but it was also already further discussed in the Supplementary Information L53-89. Specifically, we tested for each heatwave which distribution led to the best results in terms of Bayesian Information Criteria. We keep this section unchanged, which shows that the GEV is indeed the best approximation.

Statistically, the point raised by the referee is very interesting. Basically, it boils down to the following question. *Do extremes from sums (or equivalently averages) of GEV distributed block maxima better fitted by a GEV or by a Gaussian random variable?* We have compared the two fits for our dataset and the GEV appears to outperform the normal distribution. One can wonder if this is specific to this dataset or if this finding¹ is more general, i.e. a probability argument can be invoked.

In the case of extreme temperatures, most studies point towards the assumption that the shape GEV parameter ξ of annual daily maxima of temperatures is negative. So, we will mathematically treat this case². In this setup, the statistical literature is sparse and we need to develop our own reasoning here to answer the question at hand. To simplify notations³, we assume that X_1 and X_2 are standardized $GEV(0,1,\xi)$ with $\xi < 0$ and we are interested by the upper behaviour of the sum $X_1 + X_2$. As GEV distributed random variables, we have

$$P(X_1 > x) = P(X_2 > x) = 1 - \exp\left(-\overline{H}_\xi(x)\right)$$

where \overline{H}_ξ represents the survival of generalized Pareto (GPD) distribution family

$$\overline{H}_\xi(x) = P(X > x) = (1 + \xi x)_+^{-1/\xi}, \text{ with } a_+ = \max(0, a)$$

Hence, for large x near the upper bound $-1/\xi$, we can write

$$P(X_1 > x) = P(X_2 > x) \approx \overline{H}_\xi(x) \quad (1)$$

And, hence, the exceedance above u (i.e., given that $X_1 > u$, computes the excess of $X_1 - u$) denoted

$$X_1^{(u)} \equiv [X_1 - u | X_1 > u]$$

can be approximated by a GPD distribution with scale parameter $(1 + \xi u)$ and shape parameter ξ . This simply means that exceedances of GEV random variables behave as GPD in the upper tail (a well-known result in extreme value theory). Coming back to our question of the upper tail behavior of the sum of two GEV random variables, we can then write, for any $2b > x > b$ with $b = -1/\xi$ the upper bound of X_i and $u = x - b > 0$,

$$\begin{aligned} P(X_1 + X_2 > x) &= P(X_1 + X_2 > x | \min(X_1, X_2) > x - b)P(\min(X_1, X_2) > x - b), \text{ see footnote}^4 \\ &= P((X_1 - u) + (X_2 - u) > x - 2u | \min(X_1, X_2) > x - b)P(\min(X_1, X_2) > u) \\ &\approx P\left(X_1^{(u)} + X_2^{(u)} > 2b - x\right)P(\min(X_1, X_2) > u), \text{ as } x - 2u = 2b - x \text{ for } u = b - x \end{aligned}$$

In a nutshell, this allows us to rephrase our question in terms of the sum of two GPD random variables. Recently, Richards and Tawn (2022)⁸¹ studies this question in function of the dependence between $X_1^{(u)}$ and $X_2^{(u)}$. Their theorem 1 applied to $X_1^{(u)} + X_2^{(u)}$ with $\sigma_1 = \sigma_2 = 1 + \xi u = (2b - x)/b$ for negative ξ gives that:

¹ This has also been noticed in other applied D&A studies.

² The case $\xi > 0$ needs a different treatment as heavy and bounded tails are different.

³ The same reasoning can be applied for $X_1 + \dots + X_d$, with $X_i = \mu + \sigma GEV(0,1,\xi)$

⁴ Note that $\min(X_1, X_2) > x - b$ implies $X_1 + X_2 > x$, as $X_1 + X_2 = \min(X_1, X_2) + \max(X_1, X_2)$, and $\max(X_1, X_2) \leq b$

$$P\left(X_1^{(u)} + X_2^{(u)} > 2b - x\right) \approx K \left(1 + \frac{\xi}{2(1 + \xi u)}(2b - x)\right)^{\frac{-1}{\eta\xi}} = K 2^{\frac{-1}{\eta\xi}}$$

Where K represents a positive normalizing constant and η characterizes the extremal dependence between X_1 and X_2 . In particular, this later feature with η means that $P(\min(X_1, X_2) > v)$ is proportional to $\bar{H}^{1/\eta}(v)$ for large v . It follows, as $b = -1/\xi$ and $u = x - b$, that

$$P(X_1 + X_2 > x) \approx K 2^{\frac{-1}{\eta\xi}} P(\min(X_1, X_2) > x - b) \approx C \bar{H}^{1/\eta}(x - b) \quad (2)$$

To sum up, by combining this tail equivalence GEV and GPD tails and leveraging Richards and Tawn 2022)⁸¹, we have deduced that the upper tail of the sum of two dependent GEV distributed random variables is driven by Equation (2). This equation corresponds to a GPD tail, or equivalently via (1) to a GEV tail. This indicates that we should not be surprised to have an adequate fit for extremes generated by the two sum of GEV distributed random variables. Compared to a Gaussian fit, one has to recall Mill's ratio of a standard normal random variable Z that states

$$P(Z > z) \approx \frac{\phi(z)}{z} = \frac{1}{z\sqrt{2\pi}} \exp\left(-\frac{z^2}{2}\right)$$

where $\phi(\cdot)$ represents the standard normal pdf. This ratio is not an appropriate approximation of (2). This gives us a theoretical reason that may explain why GEV distribution appears to outperform Gaussian fit for extremes from sums of block maxima in our study.

Concerning our manuscript, we think that the mathematical arguments presented here are not adapted to a journal like Nature and, instead, they should be presented in a statistical journal where detailed computations, proofs and simulations could be given. But, if the editor thinks that they are needed, we could add them in the supplementary material.

Referee #4 Comment 2.1, part b

The practice of averaging temperatures across large regions during heatwave events obscures the true extremity of localized heatwaves and ignores spatial heterogeneity. For example, the 2021 Pacific Northwest heatwave affected diverse regions, including British Columbia, Washington, and Oregon, each with its own distinct topography and climatology. Simply averaging temperatures across such a large domain overlooks important local variations in heatwave intensity and the specific drivers behind the extremes in each subregion.

Response to Referee #4 Comment 2.1, part b

Working with average temperatures across large regions is common in EEA^{7,22,73,74}. Besides, the regions that we are using are based on EM-DAT and hence correspond to areas with reported impacts, and hence not large boxes centered on events as usually conducted by EEA studies. We argue that this results in a more impact-oriented heatwave representation. Finally, we point out that, as initially explained in the Supplementary Information L44-45, we average over each grid point within the boundaries specified by EM-DAT to obtain a heat indicator (average temperature) tailored to affected regions. One could argue that, e.g., by severely restricting our analysis to areas with the most anomalous temperatures, we focus on the part of heatwaves that are most extreme which is not necessarily congruent with the most affected areas (due to spatially inhomogeneous population). As a last remark, we would like to point out that the purpose is to assess the influence of climate change on a heatwave, in other words how global warming affected a regional event. While local features would modulate the

local effect of global warming, the objective remains to establish a link that holds for regional averages and the respective timescale of each event.

Referee #4 Comment 2.2

(2.2) The authors conducted 187 individual analyses, which hardly constitutes a systematic study because they ignore spatial-temporal dependence. Recent advances in statistical modeling allow for interpolating the GEV distributions between locations without spatial averaging, and also improving the analysis at individual stations by “borrowing strength” across stations while incorporating useful meteorological and topological covariates (see e.g., Cooley et al., 2007; Risser et al., 2019; Russell et al., 2020; Zhang et al., 2024). The authors would benefit from adopting these more sophisticated methods.

Response to Referee #4 Comment 2.2

For the sake of clarity, the term “systematic” characterizes here the fact that the EEA has been applied over all reported heatwaves, as opposed to individual analyses. The “ignorance” of spatial-temporal dependence does not affect the systematic characteristic of this work. As explained above, temporal dependence is correctly accounted for, and spatial averaging over the region of the events is here again a well-established element of EEA.

We would like to thank Referee #4 for suggesting these “more sophisticated methods”. While they do account for fine spatial-temporal heatwave structures, as requested in your comments (2.1), (2.2) and (2.3), these approaches are not applicable here, and hence do not serve the purpose of this manuscript.

- Missing weather stations: some regions analyzed here have too few weather stations, justifying the use of reanalysis datasets to enable a more consistent analysis. This issue hinders the use of such methods for such regions.
- Modelling of covariates by ESMs: following the statistical model of Zhang et al, 2024⁸⁰ with 6 covariates including urbanization would require using either only observations, or calculating all required covariates with ESMs as well. The resolutions of ESMs would not be adapted to calculate covariates like the urbanization. This issue reduces the applicability of such methods to model data. We would also need to assess the contributions of carbon majors to the other covariates, but this is not feasible.
- Different objectives: the studies that you suggest are indeed very interesting contributions to EEA, showing how local features modulate the influence of climate change on extreme events. Yet, our objective remains the traditional objective of EEA, that is to say the average impact of climate change over the region of the event. As such, relying on more sophisticated statistical models does not add value to our analysis, especially considering the added parametric uncertainty and arguably unnecessary model complexity given our objective.
- Weakness of such models: because these statistical models rely heavily on weather stations, only observations can be used, recorded over a relatively short time. On the contrary, models are meant to provide additional lines of evidence for an improved confidence in the results. We remind that short records affect the robustness of EEA, especially for highly improbable events such as the 2021 PNW heatwave^{3,5}.

Referee #4 Comment 2.3

(2.3) Including GMT as a covariate in the GEV model for the location parameter is problematic. While changes in GMT are partially attributable to anthropogenic climate change, they also reflect multiannual natural variability (e.g., ENSO, NAO, MJO, etc.) within the climate system.

Furthermore, using GMT as the sole covariate oversimplifies the complex drivers of heatwaves, which can involve various meteorological factors such as atmospheric blocking patterns, solar radiation, and cloud cover. For instance, the southeast and central United States have experienced a “global warming hole” with negative or insignificant temperature trends (see e.g., Mascioli et al., 2017), which contradicts the assumption that rising GMT is universally linked to increases in heatwave intensity. A more nuanced approach, accounting for both anthropogenic and natural climate variability, is needed.

Response to Referee #4 Comment 2.3

This comment is a continuation of the former comment on the use of more sophisticated models, with more covariates. We acknowledge that the Earth system is more complex than simple statistical models, that the local temperatures every year will be affected through large scale oscillations, by local effects such as albedo and aerosols, by the local water budget, etc. However, the purpose of this attribution analysis, like the majority of others, remains to assess how climate change has affected an event, thus the overall forcing. This is why we deal with natural variability driven by large scale oscillations following the WWA protocol, ie with smoothed GMST. There are indeed cases where climate change causes multiple local evolutions that compensate, leading to “warming holes”. In these cases, this would yield a near-zero slope parameter in the location function dependent on GMST, consequently event probabilities and intensities hardly change for more or less global warming, and so this also doesn’t distort the overall result. There would not be strong evolutions in frequency/intensity of the heatwaves, and a low link with GMST. To summarize, it is not a contradiction, this is simply acknowledging that sometimes, somewhere, climate change may not contribute to the heatwaves. Given the overwhelming scientific literature on the topic, we are confident that this well-established method does fit the purpose of this study.

References

- 1 Philip, S. Y. *et al.* Rapid attribution analysis of the extraordinary heat wave on the Pacific coast of the US and Canada in June 2021. *Earth Syst. Dynam.* **13**, 1689-1713 (2022). <https://doi.org/10.5194/esd-13-1689-2022>
- 2 Bartusek, S., Kornhuber, K. & Ting, M. 2021 North American heatwave amplified by climate change-driven nonlinear interactions. *Nature Climate Change* **12**, 1143-1150 (2022). <https://doi.org/10.1038/s41558-022-01520-4>
- 3 Zeder, J. & Fischer, E. M. Quantifying the statistical dependence of mid-latitude heatwave intensity and likelihood on prevalent physical drivers and climate change. *Adv. Stat. Clim. Meteorol. Oceanogr.* **9**, 83-102 (2023). <https://doi.org/10.5194/ascmo-9-83-2023>
- 4 Pons, F. M. E., Yiou, P., Jézéquel, A. & Messori, G. Simulating the Western North America heatwave of 2021 with analogue importance sampling. *Weather and Climate Extremes* **43**, 100651 (2024). <https://doi.org/https://doi.org/10.1016/j.wace.2024.100651>
- 5 Zeder, J., Sippel, S., Pasche, O. C., Engelke, S. & Fischer, E. M. The Effect of a Short Observational Record on the Statistics of Temperature Extremes. *Geophysical Research Letters* **50**, e2023GL104090 (2023). <https://doi.org/https://doi.org/10.1029/2023GL104090>
- 6 Ciavarella, A. *et al.* Prolonged Siberian heat of 2020 almost impossible without human influence. *Climatic Change* **166**, 9 (2021). <https://doi.org/10.1007/s10584-021-03052-w>
- 7 Philip, S. *et al.* A protocol for probabilistic extreme event attribution analyses. *Adv. Stat. Clim. Meteorol. Oceanogr.* **6**, 177-203 (2020). <https://doi.org/10.5194/ascmo-6-177-2020>
- 8 van Oldenborgh, G. J. *et al.* Pathways and pitfalls in extreme event attribution. *Climatic Change* **166**, 13 (2021). <https://doi.org/10.1007/s10584-021-03071-7>
- 9 Jiang, K. *et al.* Attributed radiative forcing of air pollutants from biomass and fossil burning emissions. *Environmental Pollution* **306**, 119378 (2022). <https://doi.org/https://doi.org/10.1016/j.envpol.2022.119378>
- 10 Boyd, E. *et al.* Loss and damage from climate change: A new climate justice agenda. *One Earth* **4**, 1365-1370 (2021). <https://doi.org/https://doi.org/10.1016/j.oneear.2021.09.015>
- 11 Setzer, J. & Higham, C. Global trends in climate change litigation: 2023 snapshot. (2023).
- 12 Heri, C. KlimaSeniorinnen and its discontents: Climate change at the European Court of Human Rights. *EUROPEAN HUMAN RIGHTS LAW REVIEW*, 317-331 (2024).
- 13 Stuart-Smith, R. F. *et al.* Filling the evidentiary gap in climate litigation. *Nature Climate Change* **11**, 651-655 (2021). <https://doi.org/10.1038/s41558-021-01086-7>
- 14 Clarke, B., Otto, F., Stuart-Smith, R. & Harrington, L. Extreme weather impacts of climate change: an attribution perspective. *Environmental Research: Climate* **1**, 012001 (2022). <https://doi.org/10.1088/2752-5295/ac6e7d>
- 15 Canadell, J. G. *et al.* in *Climate Change 2021: The Physical Science Basis. Contribution of Working Group I to the Sixth Assessment Report of the Intergovernmental Panel on Climate Change* (eds V. Masson-Delmotte *et al.*) 673–816 (Cambridge University Press, 2021).
- 16 Friedlingstein, P. *et al.* Global Carbon Budget 2023. *Earth Syst. Sci. Data* **15**, 5301-5369 (2023). <https://doi.org/10.5194/essd-15-5301-2023>
- 17 Franta, B. Early oil industry knowledge of CO₂ and global warming. *Nature Climate Change* **8**, 1024-1025 (2018). <https://doi.org/10.1038/s41558-018-0349-9>
- 18 Bonneuil, C., Choquet, P.-L. & Franta, B. Early warnings and emerging accountability: Total's responses to global warming, 1971–2021. *Global*

- Environmental Change* **71**, 102386 (2021).
<https://doi.org/https://doi.org/10.1016/j.gloenvcha.2021.102386>
- 19 Supran, G., Rahmstorf, S. & Oreskes, N. Assessing ExxonMobil's global warming
projections. *Science* **379**, eabk0063 (2023). <https://doi.org/10.1126/science.abk0063>
- 20 Blattner, C. E. *et al.* How science bolstered a key European climate-change case.
Nature **621**, 255-257 (2023). <https://doi.org/10.1038/d41586-023-02809-w>
- 21 Law, S. C. f. C. C. *Climate Change Litigation Databases*,
<<https://climatecasechart.com/>> (2024).
- 22 Climate Signals. *Science Sources: Detection and Attribution*,
<<https://www.climatesignals.org/reports/attribution>> (2024).
- 23 Thompson, V. *et al.* The most at-risk regions in the world for high-impact heatwaves.
Nature Communications **14**, 2152 (2023). <https://doi.org/10.1038/s41467-023-37554-1>
- 24 Leach, N. J. *et al.* Heatwave attribution based on reliable operational weather
forecasts. *Nature Communications* **15**, 4530 (2024). <https://doi.org/10.1038/s41467-024-48280-7>
- 25 Otto, F. E. L., Skeie, R. B., Fuglestedt, J. S., Berntsen, T. & Allen, M. R. Assigning
historic responsibility for extreme weather events. *Nature Climate Change* **7**, 757-759
(2017). <https://doi.org/10.1038/nclimate3419>
- 26 Ye, Y. *et al.* Attribution of a record-breaking cold event in the historically warmest
year of 2023 and assessing future risks. *npj Climate and Atmospheric Science* **8**, 14
(2025). <https://doi.org/10.1038/s41612-024-00886-w>
- 27 Tradowsky, J. S. *et al.* Attribution of the heavy rainfall events leading to severe
flooding in Western Europe during July 2021. *Climatic Change* **176**, 90 (2023).
<https://doi.org/10.1007/s10584-023-03502-7>
- 28 Arias, P. A. *et al.* Interplay between climate change and climate variability: the 2022
drought in Central South America. *Climatic Change* **177**, 6 (2023).
<https://doi.org/10.1007/s10584-023-03664-4>
- 29 Rivera, J. A. *et al.* 2022 early-summer heatwave in Southern South America: 60
times more likely due to climate change. *Climatic Change* **176**, 102 (2023).
<https://doi.org/10.1007/s10584-023-03576-3>
- 30 Li, S. & Otto, F. E. L. The role of human-induced climate change in heavy rainfall
events such as the one associated with Typhoon Hagibis. *Climatic Change* **172**, 7
(2022). <https://doi.org/10.1007/s10584-022-03344-9>
- 31 Luu, L. N. *et al.* Attribution of typhoon-induced torrential precipitation in Central
Vietnam, October 2020. *Climatic Change* **169**, 24 (2021).
<https://doi.org/10.1007/s10584-021-03261-3>
- 32 Liu, Z., Eden, J. M., Dieppois, B. & Blackett, M. A global view of observed changes in
fire weather extremes: uncertainties and attribution to climate change. *Climatic
Change* **173**, 14 (2022). <https://doi.org/10.1007/s10584-022-03409-9>
- 33 Cael, B. B., Burger, F. A., Henson, S. A., Britten, G. L. & Frölicher, T. L. Historical
and future maximum sea surface temperatures. *Science Advances* **10**, eadj5569
(2024). <https://doi.org/10.1126/sciadv.adj5569>
- 34 Morim, J. *et al.* Understanding uncertainties in contemporary and future extreme
wave events for broad-scale impact and adaptation planning. *Science Advances* **9**,
eade3170 (2023). <https://doi.org/10.1126/sciadv.ade3170>
- 35 Otto, F. E. L. *et al.* Climate change increased extreme monsoon rainfall, flooding
highly vulnerable communities in Pakistan. *Environmental Research: Climate* **2**,
025001 (2023). <https://doi.org/10.1088/2752-5295/acbfd5>
- 36 Philip, S. Y., Kew, S. F., van der Wiel, K., Wanders, N. & Jan van Oldenborgh, G.
Regional differentiation in climate change induced drought trends in the Netherlands.
Environmental Research Letters **15**, 094081 (2020). <https://doi.org/10.1088/1748-9326/ab97ca>

- 37 Qian, C. *et al.* Rapid attribution of the record-breaking heatwave event in North China in June 2023 and future risks. *Environmental Research Letters* **19**, 014028 (2024). <https://doi.org/10.1088/1748-9326/ad0dd9>
- 38 Zachariah, M. *et al.* Attribution of 2022 early-spring heatwave in India and Pakistan to climate change: lessons in assessing vulnerability and preparedness in reducing impacts. *Environmental Research: Climate* **2**, 045005 (2023). <https://doi.org/10.1088/2752-5295/acf4b6>
- 39 Harrington, L. J. *et al.* Limited role of climate change in extreme low rainfall associated with southern Madagascar food insecurity, 2019–21. *Environmental Research: Climate* **1**, 021003 (2022). <https://doi.org/10.1088/2752-5295/aca695>
- 40 Dhasmana, M. K., Mondal, A. & Zachariah, M. On the role of climate change in the 2018 flooding event in Kerala. *Environmental Research Letters* **18**, 084016 (2023). <https://doi.org/10.1088/1748-9326/ace6c0>
- 41 Dunne, K. B. J., Dee, S. G., Reinders, J., Muñoz, S. E. & Nittrouer, J. A. Examining the impact of emissions scenario on lower Mississippi River flood hazard projections. *Environmental Research Communications* **4**, 091001 (2022). <https://doi.org/10.1088/2515-7620/ac8d53>
- 42 Lott, F. C. *et al.* Quantifying the contribution of an individual to making extreme weather events more likely. *Environmental Research Letters* **16**, 104040 (2021). <https://doi.org/10.1088/1748-9326/abe9e9>
- 43 van Oldenborgh, G. J. *et al.* Attribution of the Australian bushfire risk to anthropogenic climate change. *Nat. Hazards Earth Syst. Sci.* **21**, 941-960 (2021). <https://doi.org/10.5194/nhess-21-941-2021>
- 44 Rousi, E. *et al.* The extremely hot and dry 2018 summer in central and northern Europe from a multi-faceted weather and climate perspective. *Nat. Hazards Earth Syst. Sci.* **23**, 1699-1718 (2023). <https://doi.org/10.5194/nhess-23-1699-2023>
- 45 Sippel, S. *et al.* Could an extremely cold central European winter such as 1963 happen again despite climate change? *Weather Clim. Dynam.* **5**, 943-957 (2024). <https://doi.org/10.5194/wcd-5-943-2024>
- 46 Kew, S. F. *et al.* Impact of precipitation and increasing temperatures on drought trends in eastern Africa. *Earth Syst. Dynam.* **12**, 17-35 (2021). <https://doi.org/10.5194/esd-12-17-2021>
- 47 Pietroiusti, R. *et al.* Possible role of anthropogenic climate change in the record-breaking 2020 Lake Victoria levels and floods. *Earth Syst. Dynam.* **15**, 225-264 (2024). <https://doi.org/10.5194/esd-15-225-2024>
- 48 Vautard, R. *et al.* Human influence on growing-period frosts like in early April 2021 in central France. *Nat. Hazards Earth Syst. Sci.* **23**, 1045-1058 (2023). <https://doi.org/10.5194/nhess-23-1045-2023>
- 49 Schumacher, D. L. *et al.* Detecting the human fingerprint in the summer 2022 western–central European soil drought. *Earth Syst. Dynam.* **15**, 131-154 (2024). <https://doi.org/10.5194/esd-15-131-2024>
- 50 Vautard, R. *et al.* Human influence on European winter wind storms such as those of January 2018. *Earth Syst. Dynam.* **10**, 271-286 (2019). <https://doi.org/10.5194/esd-10-271-2019>
- 51 Carrasco-Escaff, T., Garreaud, R., Bozkurt, D., Jacques-Coper, M. & Pauchard, A. The key role of extreme weather and climate change in the occurrence of exceptional fire seasons in south-central Chile. *Weather and Climate Extremes* **45**, 100716 (2024). <https://doi.org/https://doi.org/10.1016/j.wace.2024.100716>
- 52 Qian, C., Ye, Y., Bevacqua, E. & Zscheischler, J. Human influences on spatially compounding flooding and heatwave events in China and future increasing risks. *Weather and Climate Extremes* **42**, 100616 (2023). <https://doi.org/https://doi.org/10.1016/j.wace.2023.100616>
- 53 Kimutai, J., New, M., Wolski, P. & Otto, F. Attribution of the human influence on heavy rainfall associated with flooding events during the 2012, 2016, and 2018

- March-April-May seasons in Kenya. *Weather and Climate Extremes* **38**, 100529 (2022). <https://doi.org/https://doi.org/10.1016/j.wace.2022.100529>
- 54 Zachariah, M., Kumari, S., Mondal, A., Haustein, K. & Otto, F. E. L. Attribution of the 2015 drought in Marathwada, India from a multivariate perspective. *Weather and Climate Extremes* **39**, 100546 (2023). <https://doi.org/https://doi.org/10.1016/j.wace.2022.100546>
- 55 van Daalen, K. R. *et al.* The 2024 Europe report of the Lancet Countdown on health and climate change: unprecedented warming demands unprecedented action. *The Lancet Public Health* **9**, e495-e522 (2024). [https://doi.org/10.1016/S2468-2667\(24\)00055-0](https://doi.org/10.1016/S2468-2667(24)00055-0)
- 56 Hannart, A., Pearl, J., Otto, F. E. L., Naveau, P. & Ghil, M. Causal counterfactual theory for the attribution of weather and climate-related events. *Bulletin of the American Meteorological Society* **97**, 99-110 (2016). <https://doi.org/10.1175/BAMS-D-14-00034.1>
- 57 Hohne, N. *et al.* *Contributions of individual countries' emissions to climate change and their uncertainty*. Vol. 106 (2011).
- 58 Nauels, A. *et al.* Attributing long-term sea-level rise to Paris Agreement emission pledges. *Proceedings of the National Academy of Sciences* **116**, 23487-23492 (2019). <https://doi.org/10.1073/pnas.1907461116>
- 59 Fu, B. *et al.* The contributions of individual countries and regions to the global radiative forcing. *Proceedings of the National Academy of Sciences* **118**, e2018211118 (2021). <https://doi.org/10.1073/pnas.2018211118>
- 60 Li, B. *et al.* The contribution of China's emissions to global climate forcing. *Nature* **531**, 357-361 (2016). <https://doi.org/10.1038/nature17165>
- 61 Lee, D. S. *et al.* The contribution of global aviation to anthropogenic climate forcing for 2000 to 2018. *Atmospheric Environment* **244**, 117834 (2021). <https://doi.org/https://doi.org/10.1016/j.atmosenv.2020.117834>
- 62 Licker, R. *et al.* Attributing ocean acidification to major carbon producers. *Environmental Research Letters* **14**, 124060 (2019). <https://doi.org/10.1088/1748-9326/ab5abc>
- 63 Fu, B. *et al.* Climate Warming Mitigation from Nationally Determined Contributions. *Advances in Atmospheric Sciences* **39**, 1217-1228 (2022). <https://doi.org/10.1007/s00376-022-1396-8>
- 64 Beusch, L. *et al.* Responsibility of major emitters for country-level warming and extreme hot years. *Communications Earth & Environment* **3**, 7 (2022). <https://doi.org/10.1038/s43247-021-00320-6>
- 65 Dahl, K. A. *et al.* Quantifying the contribution of major carbon producers to increases in vapor pressure deficit and burned area in western US and southwestern Canadian forests. *Environmental Research Letters* **18**, 064011 (2023). <https://doi.org/10.1088/1748-9326/abce8>
- 66 Jones, M. W. *et al.* National contributions to climate change due to historical emissions of carbon dioxide, methane, and nitrous oxide since 1850. *Scientific Data* **10**, 155 (2023). <https://doi.org/10.1038/s41597-023-02041-1>
- 67 Ekwurzel, B. *et al.* The rise in global atmospheric CO₂, surface temperature, and sea level from emissions traced to major carbon producers. *Climatic Change* **144**, 579-590 (2017). <https://doi.org/10.1007/s10584-017-1978-0>
- 68 Wehner, M. Connecting extreme weather events to climate change. *Physics Today* **76**, 40-46 (2023). <https://doi.org/10.1063/pt.3.5309>
- 69 Van Oldenborgh, G. J. *et al.* Attributing and Projecting Heatwaves Is Hard: We Can Do Better. *Earth's Future* **10**, e2021EF002271 (2022). <https://doi.org/https://doi.org/10.1029/2021EF002271>
- 70 Kim, Y.-H., Min, S.-K., Zhang, X., Sillmann, J. & Sandstad, M. Evaluation of the CMIP6 multi-model ensemble for climate extreme indices. *Weather and Climate Extremes* **29**, 100269 (2020). <https://doi.org/https://doi.org/10.1016/j.wace.2020.100269>

- 71 Fan, X., Miao, C., Duan, Q., Shen, C. & Wu, Y. The Performance of CMIP6 Versus CMIP5 in Simulating Temperature Extremes Over the Global Land Surface. *Journal of Geophysical Research: Atmospheres* **125**, e2020JD033031 (2020). <https://doi.org/https://doi.org/10.1029/2020JD033031>
- 72 Otto, F. E. L. Attribution of Extreme Events to Climate Change. *Annual Review of Environment and Resources* **48**, 813-828 (2023). <https://doi.org/10.1146/annurev-environ-112621-083538>
- 73 WWA. *Event papers*, <<https://www.worldweatherattribution.org/event-papers/>> (2024).
- 74 WWA. *Analysis of heatwaves*, <<https://www.worldweatherattribution.org/analysis/heatwave/>> (2024).
- 75 Perkins-Kirkpatrick, S. E. *et al.* Frontiers in attributing climate extremes and associated impacts. *Frontiers in Climate* **6** (2024).
- 76 Russo, E. & Domeisen, D. I. V. Increasing Intensity of Extreme Heatwaves: The Crucial Role of Metrics. *Geophysical Research Letters* **50**, e2023GL103540 (2023). <https://doi.org/https://doi.org/10.1029/2023GL103540>
- 77 Vicedo-Cabrera, A. M. *et al.* The burden of heat-related mortality attributable to recent human-induced climate change. *Nature Climate Change* **11**, 492-500 (2021). <https://doi.org/10.1038/s41558-021-01058-x>
- 78 Gasparrini, A. *et al.* Projections of temperature-related excess mortality under climate change scenarios. *The Lancet Planetary Health* **1**, e360-e367 (2017). [https://doi.org/10.1016/S2542-5196\(17\)30156-0](https://doi.org/10.1016/S2542-5196(17)30156-0)
- 79 Ballester, J. *et al.* Heat-related mortality in Europe during the summer of 2022. *Nature Medicine* **29**, 1857-1866 (2023). <https://doi.org/10.1038/s41591-023-02419-z>
- 80 Zhang, L., Risser, M. D., Wehner, M. F. & O'Brien, T. A. Leveraging Extremal Dependence to Better Characterize the 2021 Pacific Northwest Heatwave. *Journal of Agricultural, Biological and Environmental Statistics* (2024). <https://doi.org/10.1007/s13253-024-00636-8>
- 81 Richards, J. & Tawn, J. A. On the tail behaviour of aggregated random variables. *Journal of Multivariate Analysis* **192**, 105065 (2022). <https://doi.org/https://doi.org/10.1016/j.jmva.2022.105065>

Response to Referees for the manuscript

Systematic attribution of heatwaves to the emissions of carbon majors

Yann Quilcaille¹, Lukas Gudmundsson¹, Dominik L. Schumacher¹, Thomas Gasser², Richard Heede³, Corina Heri⁴, Quentin Lejeune⁵, Shruti Nath⁶, Philippe Naveau⁷, Wim Thiery⁸, Carl-Friedrich Schleussner^{2,9}, Sonia I. Seneviratne¹

¹Institute for Atmospheric and Climate Science, Department of Environmental Systems Science, ETH Zurich, Zurich, Switzerland.

²International Institute for Applied Systems Analysis (IIASA), Laxenburg, Austria.

³Climate Accountability Institute, Snowmass, Colorado, USA

⁴Department of Public Law and Governance, Tilburg University, Tilburg, Netherlands

⁵Climate Analytics, Berlin, Germany

⁶Atmospheric, Oceanic and Planetary Physics, Department of Physics, University of Oxford

⁷Laboratoire des Sciences du Climat et de l'Environnement, ESTIMR, CNRS-CEA-UVSQ, Gif-sur-Yvette, France

⁸Vrije Universiteit Brussel, Department of Water and Climate, Brussels, Belgium.

⁹Integrative Research Institute on Transformations of Human-Environment Systems (IRI THESys) and the Geography Department, Humboldt-Universität zu Berlin, Berlin, Germany.

Correspondence to: Yann Quilcaille (yann.quilcaille@env.ethz.ch)

We would like to thank the three Referees for their constructive and insightful feedback. We have carefully integrated all comments, and the manuscript has become clearer and even more robust, thus improving the quality of this manuscript. Here is a summary of the modifications:

- Clarifications regarding the definitions of the events, following Referee 1's suggestions.
- Addition of goodness-of-fit tests for all events, following Referee 3's recommendations. All but 9 heatwaves had sufficient performance, while the 9 others were removed from analysis.
- Addition of non-linear Granger-causality tests, building on Referee 3's recommendations. For all but 3 heatwaves, GMST is Granger-causally related to the respective heatwave indicator. The 3 other events were removed from analysis.
- Addition of 2023 heatwaves to the manuscript. A recent update to the Carbon Majors database enabled an extension of the analysis from initially 187 to 226 heatwaves (not counting those removed through the aforementioned tests).
- Clarifications in the structure of the Methods and in the Supplementary Information for better navigation by the reader.
- Update of GitHub & Zenodo repositories.

Contents

Response to Referee #1	3
Response to Referee #2	5
Response to Referee #5	6
References.....	11

Response to Referee #1

Referee #1 Comment 1

The article conducts a systematic event attribution of 187 historical heatwaves, and attributes the effects to individual carbon majors. This is my second review of the article and I thank that authors for considering all my comments and find the current version much improved, particularly regarding where this present paper relates to existing literature.

Response to Referee #1 Comment 1

We want to thank Referee #1 for this perspective on our modifications. We will do our best to integrate all ensuing comments.

Referee #1 Comment 2

The major remaining concern I have regards the event definition. What exactly are the black dots on figure 1? From reading the text and caption my understanding is that they are Tmax averaged over the region and period of the event – with one value per year. I find this quite a strange choice and do not understand why have you chosen this definition rather than say annual maximum values averaged over the length of the event which I think are the more typical and robust approaches?

Indeed, the studies that you compare to for the Pacific NW event use annual maximum Tmax (TXx) and a 5-year block maximum of annual maximum seven-day running mean air temperature.

Could you clarify and justify your choice of approach, particularly given that you are using a GEV fit which as I understand it is only applicable to fit to distributions of extreme values. The choice of how to define an event is a key part of event attribution so I think this point needs more detail in the main text.

Response to Referee #1 Comment 2

Thank you for raising this concern. We fully agree that the definition of the indicator is a crucial step in the process of Extreme Event Attribution. Our choice of indicator is based on the reporting of the heatwave to EM-DAT. For instance, the Pacific NW event has been reported separately to EM-DAT for both Canada and the USA. In panel b of Figure 1, we show the reporting on the USA. According to the reporting to EM-DAT, the heatwave occurred over the 26-30.06.2021. Based on this reporting, we calculate the average over this period. We agree that other indicators are possible: the maximum of the daily maximum, the maximum of the daily average, the maximum of a running average, and many others¹. Our choice is motivated by the relevance of the definition for the impacts perceived by the population, which is one of the approaches proposed by the World Weather Attribution (see section 3.5 of the protocol²). For instance, using daily average accounts for the cooling over night, while daily maximum doesn't. Using average over the period of the heatwave rather than the maximum over the period provides insights on how sustained was the heatwave. Both aspects matter for impacts. We acknowledge that this justification was lacking, and we have modified the manuscript (L95-101) accordingly:

Here, the heatwave is defined using the exact spatial characterization in the EM-DAT database, as it represents how the disaster was experienced by the local populations² (Figure 1b). Daily temperatures are averaged over this period and region for every year of available observations^{3,4}. While many indicators could be used to characterize the heatwave¹, the choice

of the average over the period is motivated by its relevance for the reported impact rather than its meteorological rarity (see Supplementary Information).

Additionally, we have also detailed further this aspect in the relevant section of Supplementary Information (L62-67):

In particular, this choice is motivated by its relevance for the reported impact rather than its meteorological rarity. Heatwaves impact local populations not only through daily maximum temperatures, but also through lack of cooling over nights, which can be estimated using daily average temperatures. Sustaining high temperatures over time modify as well the impact of a heatwave, justifying the use of averages over the period of the heatwave rather than its peak.

Referee #1 Comment 3

Lines 97-101: “Using the sole” would be clearer as “using only”. The sentence following this seems a bit repetitive in terms of phrasing and it would be good to rewrite this.

Response to Referee #1 Comment 3

Thanks for this comment. We have modified the manuscript as follows (L110-115):

Using only ERA5, the Pacific Northwest heatwave of 2021 over the USA had climate change increasing its intensity by 4.4 °C compared to 1850-1900, with a 95% confidence interval from 2.2 to 6.8 °C. Adding all other datasets relevant for the region decreases the influence of climate change to a change in intensity of the Pacific Northwest heatwave of 2021 over the USA of 3.1 °C (1.4 to 5.1).

Referee #1 Comment 4

Line 139. What does “best” mean in this context?

Response to Referee #1 Comment 4

Thank you for pointing this out. In this context, it means “median” estimate when accounting for uncertainties. We have replaced “best” with “median” in two instances, L155-156:

With reference to 1850-1900, the median estimates for the changes in intensity range across events from +0.3 °C to +3.1 °C.

And L166-168

However, the median estimates show that climate change has made 55 heatwaves out of 213 (26%) at least 10,000 times more likely, which is to say virtually impossible without anthropogenic influence.

Response to Referee #2

Referee #2 Comment 1

I finally managed to go through all the referee comments and responses, with apologies for the delay in doing so.

I am impressed by the rigour and detailedness of the responses by the author team. Also, having read the critical review of Referee #4, I can absolutely understand why the authors challenged the decision to reject the manuscript. I have to say, most of their (the referee's) points were either based in misunderstandings or a general unwillingness to accept standard methods in EEA. In particular some of the comments made on the event definition by referee #4 border on straw man arguments. I admire the calm and considered response by the authors and the massive amount of extra work they invested to address their critique exhaustively.

The other two referee comments were addressed with equal care. Needless to say, my initial points of criticism have all been addressed and are now appropriately amended in the manuscript. Very well done! Therefore, I happily accept the paper as is and wholeheartedly recommend publication in Nature.

Response to Referee #2 Comment 1

I and the co-authors would like to sincerely and deeply thank Referee #2 for this positive feedback and support. Of course, should any new point arise, we will gladly account for them.

Response to Referee #5

Referee #5 Comment 1

Dear authors,

I have been asked to assess your replies to the now-absent referee #4 (R4). Upon reading the comments from the previous R4 and enough of the manuscript to grasp the core of your motivation, methods, and results, my overall feeling is that R4 was overly (and unnecessarily) critical of your manuscript and statistical methodology. I would agree with R4 that the methods used here are not necessarily state-of-the-art in the statistics literature; however, I also agree with the authors that some balance between sophistication and practicality/reproducibility is needed for large-scale studies such as this one. Ultimately, I think you have graciously revised your manuscript in light of R4 feedback while respectfully standing your ground in areas where R4 was (in my opinion) unreasonable.

Response to Referee #5 Comment 1

We would like to thank you for this pondered review of this manuscript and the replies.

Referee #5 Comment 2

1. R4 Comment 2.1(a)

Here, R4 was asking about the appropriateness of using GEV theory for analyzing block maxima from finite data while extrapolating very far into the upper tail (in some cases, to >10,000-year events). They correctly raise the point that the measurements from which you take a block maximum (daily T within a year) are definitively neither independent nor identically distributed. R4 then asks you to conduct goodness-of-fit tests to ensure appropriateness of the GEV. In your revision, the Supplementary Information now states:

“Besides this GEV model, we have also tested three other distributions with linear and non-linear evolutions of the parameters: Gaussian, skew normal and generalized Pareto. Overall, our comparisons assessed through quantile-quantile plots clearly indicate that the GEV performs the best among the four distributions, especially in terms of upper tail behaviours.”

While it is nice to know that the GEV provides the *best* fit from among the other candidates, this does not imply that the GEV does indeed provide a *good* or *appropriate* fit to the data (while the GEV may be best, it could still provide a poor, albeit relatively good, fit).

Related, in the Supplementary Materials you also state:

“The choice of including or not the event when estimating the statistical model has been extensively discussed, although no final consensus was reached^{7,16,17}. The results presented in this paper have been obtained by estimating with the event, to prevent removing points from the observational record. To ensure numerical convergence, a minimum probability of $10e-9$ was set for each point of the full sample. It implies that the attributed events under factual conditions will not have return periods higher than a billion years, which we consider long enough.”

I concur that there is no final consensus on whether the event of interest should be considered

in- or out-of-sample for the analysis. However, whether or not it is included can have major implications for the associated return periods and probabilities. As you probably know, this issue reared its ugly head in the PNW heatwave. Philip et al. (2022) directly found that treating the 2021 temperature as out-of-sample yielded an infinite return interval for the event; hence, they elected to include the event in their analysis to yield a finite return interval (and, therefore, very different return intervals with and without the event of interest). Bercos-Hickey et al. (2022) directly looked at goodness-of-fit and found that for many of the stations considered the GEV distribution did *not* provide a good fit to the data when the 2021 measurement was included, but when it was excluded, the GEV distribution was acceptable.

In my opinion, you need to consider this goodness-of-fit question more formally, particularly since (1) you include the heatwave events as in-sample and (2) you use the GEV fits to extrapolate far into the upper right tail. I'd suggest something along the lines of what was done in Bercos-Hickey et al. (2022); also, Risser et al. (2025) <https://doi.org/10.1016/j.wace.2025.100743> propose methods for conducting these checks (see their Section 3.1 and also Section 2.2.5 of the supporting information).

Response to Referee #5 Comment 2

Thank you very much for this insightful comment. Indeed, the fit of a non-stationary GEV remains a statistical model, that requires an adequate validation, even if well-established. Although we had shown that it was the best one, we acknowledge the need to show its goodness-of-fit.

We have followed the method from Risser et al, 2025 that you recommend for every heatwave and every dataset that we use for the attribution analysis⁵. More precisely, we have projected the sample onto a $GEV(0,1,\xi)$, deduced its observed quantiles for comparison to the theoretical quantiles of this GEV. Importantly, we flag that all parameters (location, scale and shape) of the non-stationary distribution have uncertainty, hence a range both in the quantiles of the sample and a range in the theoretical quantiles. In Figure 1 of this Response to the Referees, we illustrate this process with the QQ-plot for the PNW 2021 heatwave with the ERA5 dataset. Although the median of the quantiles of the sample seem slightly too low then slightly too high, the identity line remains always within the uncertainty range.

Figure 1: quantiles-quantiles plot for the PNW 2021 heatwave based on ERA5 data. Following Risser et al, 2025, we have then counted how often the quantiles of the sample not consistent with the theoretical quantiles. Importantly, we focused on the quantiles above the

median to ensure the consistency for heatwaves. For every heatwave, we have then analyzed whether the conditional distributions trained for the datasets used for analysis had less than 5% of their sample outside the confidence band. These results are summarized in Figure 2, with boxplots representing the spread over the datasets. Out of the 226 heatwaves, we remove the 9 heatwaves for which more than half of the goodness-of-fit is above the threshold at 5%. We notice that 8 out of 9 removed events occurred in India, the last one being a heatwave in Japan in 2019. We ignore why 8 out of the 14 heatwaves in India had lower performances, the 6 others had overall satisfying goodness-of-fits. After careful search, we could not pinpoint any specific reason, whether year, season, duration, region or intensity of the heatwave. Further research would be needed to identify potential reasons, for instance through the regional drivers of these specific events, but it would be in a different paper.

These results have been added in a new section in Supplementary Information. In the main text, these tests are now mentioned to justify that several heatwaves are removed from the whole analysis (L132-140):

Additional tests are conducted to assess the adequacy of the method for each event. The goodness-of-fit is assessed, validating 217 out of the 226 heatwaves, while the remaining 9 are removed from the ensuing analysis. [...] All details on these tests are provided in Supplementary information.

For information, one of the discarded events was shown in panel (d) of Figure 1 and now Figure S4. This Indian heatwave was replaced with another Indian heatwave. Besides, all the analysis in the manuscript has been modified to account for the removal of these events,

Referee #5 Comment 3

2. R4 Comment 1.5

I think the reviewer is asking about something else here, but I concur with the overall sentiment regarding the importance of clearly stated causal assumptions. I find your approach for isolating the carbon majors (averaging the ABO and AON fits) both novel and interesting, and Figure S2 is compelling. However, particularly for the observational data sources, what you

are doing (regressing one time series on another with GEV errors) is still subject to problems associated with hidden covariates (i.e., you find correlation and not causation). In some ways, your approach could be framed as Granger causality (Granger, 1969 <https://doi.org/10.2307/1912791>; see also Risser et al., 2025 <https://doi.org/10.1088/2752-5295/add046>), although the nonstationarity of both the covariates and response variables are problematic. Whether you are inferring causation or correlation will be very important if the results are ever to be used in court for loss and damages! Therefore, I think you need to be *very* clear about the nature of the attribution statements you're making, and what sort of causality is implied.

Response to Referee #5 Comment 3

We would like to thank you for this judicious comment. We acknowledge that the method of non-stationary distributions relies on correlations. Even if there is strong physical basis for causation, there may be additional effects, e.g. hidden covariates as you mention. Given that this manuscript aims at quantifying the causal chain, we have implemented Granger causality for all heatwaves to assess whether the evolution of GMST does Granger-cause the evolution of T , the indicator for the heatwave. The paragraphs that follows are only a summary of the work conducted, full details on the technical implementation and results are provided in a new section in Supplementary Information.

- Our first approach relies on the sources that you provided^{6,7}, with Granger-causation used classical differentiating variables. In other words, this test was assessing whether interannual variability in GMST can predict the interannual variability in T . We could reject the null hypothesis only for about 1/4 of the cases, i.e. proving that there is indeed causality. However, this classical approach removes the trends from GMST and T , which are essential information.
- Our second approach relies on a generalization introduced in Granger and Newbold, 1974⁸. This method introduces the effects of trends, thus testing whether short and long-term evolutions of GMST may predict the evolutions of T . With this more sophisticated test, we could show that the evolutions of GMST was causing the evolutions of T in about 2/3 of the cases. However, as pointed out in Risser et al, 2025⁷, the non-linearity of the Earth system represents a limit for Granger-causality.
- Our third approach relies on a non-linear Granger-causality framework based on machine learning^{9,10}. Using Random Forest models, we show that 214/217 of the heatwaves have the median of their test below 0.05. Two other events have 0.06, the last one has 0.11. We decide to remove these three events from the ensuing analysis, to guarantee that we have proven causation.

While all details are provided in Supplementary Information, we summarized these findings in the Main Text (L134-140) as follows:

Furthermore, although there are strong physical justifications that GMST has a causal link to the heatwave^{2,11,12}, this statistical model does not necessarily imply statistical causation. Thus we also infer the non-linear Granger causality^{7,10}. For 214 out of the 217 heatwaves, we prove with more than 95% certainty that GMST is Granger-causing the indicator of the heatwave. The 3 other events are removed from the ensuing analysis. [...] All details on these tests are provided in Supplementary information.

References

- 1 Russo, E. & Domeisen, D. I. V. Increasing Intensity of Extreme Heatwaves: The Crucial Role of Metrics. *Geophysical Research Letters* **50**, e2023GL103540 (2023). <https://doi.org/https://doi.org/10.1029/2023GL103540>
- 2 Philip, S. *et al.* A protocol for probabilistic extreme event attribution analyses. *Adv. Stat. Clim. Meteorol. Oceanogr.* **6**, 177-203 (2020). <https://doi.org/10.5194/ascmo-6-177-2020>
- 3 Hersbach, H. *et al.* The ERA5 global reanalysis. *Quarterly Journal of the Royal Meteorological Society* **146**, 1999-2049 (2020). <https://doi.org/https://doi.org/10.1002/qj.3803>
- 4 Bell, B. *et al.* The ERA5 global reanalysis: Preliminary extension to 1950. *Quarterly Journal of the Royal Meteorological Society* **147**, 4186-4227 (2021). <https://doi.org/https://doi.org/10.1002/qj.4174>
- 5 Risser, M. D., Zhang, L. & Wehner, M. F. Data-driven upper bounds and event attribution for unprecedented heatwaves. *Weather and Climate Extremes* **47**, 100743 (2025). <https://doi.org/https://doi.org/10.1016/j.wace.2025.100743>
- 6 Granger, C. W. J. Investigating Causal Relations by Econometric Models and Cross-spectral Methods. *Econometrica* **37**, 424-438 (1969). <https://doi.org/10.2307/1912791>
- 7 Risser, M. D., Ombadi, M. & Wehner, M. F. Granger causal inference for climate change attribution. *Environmental Research: Climate* **4**, 022001 (2025). <https://doi.org/10.1088/2752-5295/add046>
- 8 Granger, C. W. J. & Newbold, P. Spurious regressions in econometrics. *Journal of Econometrics* **2**, 111-120 (1974). [https://doi.org/https://doi.org/10.1016/0304-4076\(74\)90034-7](https://doi.org/https://doi.org/10.1016/0304-4076(74)90034-7)
- 9 Papagiannopoulou, C. *et al.* A non-linear Granger-causality framework to investigate climate–vegetation dynamics. *Geosci. Model Dev.* **10**, 1945-1960 (2017). <https://doi.org/10.5194/gmd-10-1945-2017>
- 10 Leng, S., Xu, Z. & Ma, H. Reconstructing directional causal networks with random forest: Causality meeting machine learning. *Chaos: An Interdisciplinary Journal of Nonlinear Science* **29** (2019). <https://doi.org/10.1063/1.5120778>
- 11 Seneviratne, S. I. *et al.* in *Climate Change 2021: The Physical Science Basis. Contribution of Working Group I to the Sixth Assessment Report of the Intergovernmental Panel on Climate Change* (eds V. Masson-Delmotte *et al.*) Ch. 11, (Cambridge University Press, 2021).
- 12 van Oldenborgh, G. J. *et al.* Pathways and pitfalls in extreme event attribution. *Climatic Change* **166**, 13 (2021). <https://doi.org/10.1007/s10584-021-03071-7>

Referee report on Nature-2024-07-15234:
“*Systematic attribution of heatwaves to the emissions of carbon majors*”

The manuscript presents an analysis of 187 historical heatwaves from 2000–2022 and attempts to attribute these events to carbon emitters and anthropogenic climate change. While the topic is highly relevant and timely, the statistical methodologies employed raise several critical issues, particularly concerning accuracy, rigor, and adherence to fundamental principles of probability theory and extreme event attribution. In its current form, the conclusions drawn from the analysis appear unreliable due to incorrect model assumptions and flawed statistical methods.

For the reasons detailed below, I recommend rejecting the manuscript. A substantial revision of the methodology and analyses is necessary before considering publication in any outlet, especially a journal as prominent as *Nature*.

1 Attribution of heatwaves to carbon majors

The core equation used to attribute heatwaves to the emissions of carbon majors is Equation (2) on Line 373:

$$R = N \cup O \cup \bigcup_{i=1, \dots, n} E_i,$$

where R denotes the factual conditions, N is the unperturbed conditions (i.e., pre-industrial/counterfactual conditions), E_i , $i = 1, \dots, n$ denote the activities of all carbon emission entities considered, and O denotes all other anthropogenic effects. However, **this equation suffers from critical logical flaws and violates the basic principles of set theory and probability theory**:

- (1) The treatment of events and probability is very not rigorous. What is the probability space S under consideration? What is its associated σ -algebra \mathcal{B} ?

The definitions of R and N as factual and counterfactual conditions, respectively, are imprecise. In probability theory, if N represents “world that might have been” conditions that never happened, $P(N)$ should be 0, while $P(R) = 1$, as the factual conditions did occur. The assumption of $\frac{P(N)}{P(R)} = 1$ (see Line 176 of the Supplementary Material)

(Line 176 of the Supplementary Material) is inconsistent with basic probability principles.

- (2) In probability theory, if we have two events N and R in a probability space (S, \mathcal{B}) such that $N \subset R$, then any occurrence $s \in N$ will satisfy $s \in R$, which means R must happen when N happens.

However, by definition, the counterfactual/unperturbed conditions N cannot happen simultaneously with the factual conditions R , which is the basis principle of extreme event attribution or causal inference. That is to say, $R \cap N = \emptyset$, which directly contradicts Equation (2) on Line 373.

- (3) A union of two events $E_1 \cup E_2$ indicates either E_1 or E_2 happens, not requiring both events to happen at the same time. However, for the factual conditions R to occur, all anthropogenic activities, including O and E_i , $i = 1, \dots, n$. Therefore, should co-occur. Thus, the correct formulation of R should involve intersections:

$$R = O \cap \bigcap_{i=1, \dots, n} E_i.$$

- (4) To derive $\delta PR_{X, E_i}$, the Equation (12) in the Supplementary Material requires simulating and computing

$$P(X | R_I), \text{ where } R_I = \bigcap_{i \in I} E_i, I \subset \{1, \dots, n + 2\}.$$

That is, one needs to run an earth system model (ESM) $2^n - 1$ times (which is equal to the number of subsets for $\{1, \dots, n\}$) to evaluate the probabilities $P(X | R_I)$ under all index sets. Considering there are $n = 122$ carbon majors under consideration in this paper, running the OSCARv3.3 driven by $2^{122} - 1$ different counterfactual scenarios is computationally infeasible. The authors should clarify how they managed this massive computational burden.

- (5) Clear communication of causal assumptions and the interpretation of conclusions is essential in climate change detection and attribution studies. Attribution results based on Earth System Models (ESMs) are inherently subject to structural uncertainties, and their reliability depends heavily on the model's ability to accurately simulate the relevant variables of interest (Wehner, 2023).

For instance, in the case of the June 26–29, 2021 Pacific Northwest (PNW) heatwave, Bercos-Hickey et al. (2022) employed ensembles of **regional** climate models, including the Weather Research and Forecasting (WRF) model and the International Centre for Theoretical Physics Regional Climate Model (RegCM 4.9.5). These simulations, initialized on June 22, 2021, used historical boundary conditions from the Global Forecast System (GFS) analysis and ran continuously through July 2, 2021. Despite the use of regional models specifically designed to capture *localized* climate dynamics, the highest

temperatures simulated by WRF were still approximately $\sim 1^\circ$ C lower than observed in the PNW.

Given this, it raises doubts as to whether a reduced-complexity **global** ESM like OSCARv3.3 can adequately reproduce the intensity of all 187 heatwave events analyzed in the study. Indeed, previous research has shown that global climate models frequently underestimate the intensity of extreme heat events (see e.g., Stott et al., 2016), further questioning the robustness of the model-based conclusions in this paper.

2 Attribution of heatwaves to climate change

To study the effect of anthropogenic climate change on the intensity of a certain heatwave event, the authors took a spatial average of the daily temperature within the impact region of the heatwave to obtain an annual time series. Then they assumed the spatially-averaged temperatures subject to a Generalized Extreme Value Distribution (GEV) distribution.

They claimed that this type of analysis applied individually to 187 different heatwave events is systematic, which I can hardly agree with. Also, **their practice violates the classic Extreme Value Theory (EVT), and also the inclusion of Global mean temperature (GMT) as a covariate in the location parameter of the GEV distribution is somewhat naïve.**

1. The univariate EVT states that given a sequence of independent random variables Y_1, Y_2, \dots with common distribution F , and let $M_n = \max\{Y_1, \dots, Y_n\}$, there exist sequences of constants $a_n > 0$ and $b_n \in \mathbb{R}$ such that

$$\frac{M_n - b_n}{a_n} \xrightarrow{d} Z, \text{ as } n \rightarrow \infty, \quad (1)$$

where Z is a non-degenerate random variable that has a GEV distribution and \xrightarrow{d} denotes convergence in distribution. See Coles (2001) for an introductory exposition.

For the GEV assumptions (1) to hold, one has to work with block maxima with a block size $n \rightarrow \infty$, which is why Statisticians usually fit GEV distributions to annual maxima or seasonal maxima. Also, it is important to make sure that the observations within a block $\{Y_1, \dots, Y_n\}$ all subject to the same distribution. Considering the daily temperature in a year has autocorrelation and seasonality among days, the records are not identically distributed. The authors must conduct a GEV goodness-of-fit test to their spatially-averaged records because the GEV assumptions have all been seriously violated.

The practice of averaging temperatures across large regions during heatwave events obscures the true extremity of localized heatwaves and ignores spatial heterogeneity. For example, the 2021 Pacific Northwest heatwave affected diverse regions, including British Columbia, Washington, and Oregon, each with its own distinct topography and climatology. Simply averaging temperatures across such a large domain overlooks important local variations in heatwave intensity and the specific drivers behind the extremes in each subregion.

2. The authors conducted 187 individual analyses, which hardly constitutes a systematic study because they ignore spatial-temporal dependence. Recent advances in statistical modeling allow for interpolating the GEV distributions between locations without spatial averaging, and also improving the analysis at individual stations by “borrowing strength” across stations while incorporating useful meteorological and topological covariates (see e.g., Cooley et al., 2007; Risser et al., 2019; Russell et al., 2020; Zhang et al., 2024). The authors would benefit from adopting these more sophisticated methods.
3. Including GMT as a covariate in the GEV model for the location parameter is problematic. While changes in GMT are partially attributable to anthropogenic climate change, they also reflect multiannual natural variability (e.g., ENSO, NAO, MJO, etc.) within the climate system. Furthermore, using GMT as the sole covariate oversimplifies the complex drivers of heatwaves, which can involve various meteorological factors such as atmospheric blocking patterns, solar radiation, and cloud cover. For instance, the southeast and central United States have experienced a “global warming hole” with negative or insignificant temperature trends (see e.g., Mascioli et al., 2017), which contradicts the assumption that rising GMT is universally linked to increases in heatwave intensity. A more nuanced approach, accounting for both anthropogenic and natural climate variability, is needed.

References

- Emily Bercos-Hickey, Travis A O’Brien, Michael F Wehner, Likun Zhang, Christina M Patricola, Huanping Huang, and Mark D Risser. Anthropogenic contributions to the 2021 pacific northwest heatwave. *Geophysical Research Letters*, 49(23):e2022GL099396, 2022.
- Stuart Coles. *An introduction to statistical modeling of extreme values*. Springer, 2001.
- Daniel Cooley, Douglas Nychka, and Philippe Naveau. Bayesian spatial modeling of extreme precipitation return levels. *Journal of the American Statistical Association*, 102(479):824–840, 2007.
- Nora R Mascioli, Michael Previdi, Arlene M Fiore, and Mingfang Ting. Timing and seasonality of the united states ‘warming hole’. *Environmental Research Letters*, 12(3):034008, 2017.
- Mark D Risser, Christopher J Paciorek, Michael F Wehner, Travis A O’Brien, and William D Collins. A probabilistic gridded product for daily precipitation extremes over the united states. *Climate Dynamics*, 53(5):2517–2538, 2019.

Brook T Russell, Mark D Risser, Richard L Smith, and Kenneth E Kunkel. Investigating the association between late spring gulf of mexico sea surface temperatures and us gulf coast precipitation extremes with focus on hurricane harvey. *Environmetrics*, 31(2): e2595, 2020.

Peter A Stott, Nikolaos Christidis, Friederike EL Otto, Ying Sun, Jean-Paul Vanderlinden, Geert Jan van Oldenborgh, Robert Vautard, Hans von Storch, Peter Walton, Pascal Yiou, et al. Attribution of extreme weather and climate-related events. *Wiley Interdisciplinary Reviews: Climate Change*, 7(1):23–41, 2016.

Michael Wehner. Connecting extreme weather events to climate change. *Physics Today*, 76(9):40–46, 2023.

Likun Zhang, Mark D Risser, Michael F Wehner, and Travis A O'Brien. Leveraging extremal dependence to better characterize the 2021 pacific northwest heatwave. *Journal of Agricultural, Biological and Environmental Statistics*, pages 1–22, 2024.